# Molecular evolution of $CO_2$-sensing ab1C neurons underlies divergent sensory responses in the *Drosophila suzukii* species group

Alice Gadau[1], Sasha Mills[1], Xin Yu Zhu Jiang[1], Cong Li[1], Nicolas Svetec[1], Ziyu Xu[1], Wanhe Li[2], Katherine I. Nagel[3]*, Li Zhao[1]*

**1** Laboratory of Evolutionary Genetics and Genomics, The Rockefeller University, New York, New York, United States of America, **2** Department of Biology, Texas A&M University, College Station, Texas, United States of America, **3** Neuroscience Institute, NYU Medical Center, New York, New York, United States of America

* katherine.nagel@nyumc.org (KIN); lzhao@rockefeller.edu (LZ)

## Abstract

Organisms evolve behavioral and morphological traits to adapt to their ecological niches, yet the genetic basis of adaptation remains largely unknown. *Drosophila suzukii* has evolved a distinctive oviposition preference for ripe fruit, unlike most *Drosophila* species such as *D. melanogaster*, which prefer overripe fruit. Carbon dioxide ($CO_2$), a metabolic volatile that increases as fruit ripens and decays, may act as a critical ecological cue shaping these preferences. Here, we focus on *D. suzukii* and its sister species *D. subpulchrella*, which shows an intermediate preference, to investigate the genetic basis of $CO_2$ responses. We report a previously unrecognized shift in $CO_2$-guided oviposition: *D. suzukii* and *D. subpulchrella* readily lay eggs on $CO_2$-enriched substrates, unlike the strong aversion displayed by *D. melanogaster*. Electrophysiological recordings revealed a species-specific sensory tuning, characterized by an early spike in $CO_2$-evoked neuronal firing in *D. suzukii* and *D. subpulchrella*—a temporal response feature absent in *D. melanogaster*. To dissect the genetic basis of this shift, we generated transgenic *D. melanogaster* expressing either the *D. suzukii* Gr63a coding sequence or the *D. subpulchrella* Gr63a *cis*-regulatory element. Remarkably, both manipulations reproduced the early-onset firing pattern of $CO_2$ sensitivity, demonstrating that either receptor function or expression can independently drive this sensitivity adaptation. Our findings reveal that evolution can shape ecological adaptation through distinct genetic mechanisms, leading to convergent physiological traits among closely related species.

which permits unrestricted use, distribution, and reproduction in any medium, provided the original author and source are credited.

**Data availability statement:** Data and code availability The data and codes used to generate figures can be found at https://github.com/LiZhaoLab/ suzukii_subpuchrella_CO2_egglaying.

**Funding:** This work was supported by National Institutes of Health (NIH) MIRA R35GM133780, the Robertson Foundation, a Vallee Scholar Program (VS-2020-35), and an Allen Distinguished Investigator Award from Paul G. Allen Family Foundation to L.Z. A.G. work was partly funded by the Rosemary Grant Advanced award from Society for the Study of Evolution and NIH NRSA T32 training grant GM066699. The funders had no role in study design, data collection and analysis, decision to publish, or preparation of the manuscript.

**Competing interests:** The authors have declared that no competing interests exist.

## Author summary

Animals rely on their senses to locate food sources and identify suitable reproductive sites in their environment. Closely related species can evolve strikingly different preferences as they adapt to new environments. For example, the invasive fruit fly *D. suzukii* lays its eggs in ripe fruit, unlike most other fruit flies, such as *D. melanogaster*, which prefer decaying fruit. Because $CO_2$ levels increase as fruit ripens and ferments, changes in how flies detect $CO_2$ may have contributed to these ecological differences. We compared $CO_2$ responses between *D. suzukii* and its sister species *D. subpulchrella*, and found that both species respond to $CO_2$ differently from *D. melanogaster*: both in their oviposition preferences and neural $CO_2$ sensitivity. By introducing either the *D. subpulchrella* or *D. suzukii* $CO_2$ receptor gene coding sequences or regulatory regions into *D. melanogaster*, we found that this altered sensitivity can arise from changes either in the receptor's protein-coding region or in the DNA elements that control its expression. Our results show that evolution can act through multiple genetic mechanisms to fine-tune sensory systems, revealing how subtle molecular changes can generate ecological diversity among closely related species.

## Introduction

*D. suzukii*, originally documented as a new species in the 1930s in East Asia [1–3], has evolved a distinct preference for ovipositing in ripening stone fruits and berries [4,5]. This novel behavior distinguishes *D. suzukii* from most *Drosophila* species, such as *D. melanogaster*, which prefer to oviposit in rotting fruits. In the 1980s, *D. suzukii* was discovered to have invaded Hawaii [6] and the North American mainland by the early 2000s [5,7]. Since its initial invasion, *D. suzukii* has been reported in over 50 countries [7] and has caused significant global crop damage as larvae hatch within ripening fruits and cause premature fruit spoilage [8–12]. *D. suzukii*'s novel egg-laying behavior is also facilitated by an enlarged ovipositor that is adorned with large pegs that form a serrated edge [13,14]. This adaptation, absent in most *Drosophila* species, enables *D. suzukii* to pierce the tougher skin of ripening fruits.

 *D. suzukii's* sister species*, D. subpulchrella,* also possesses an enlarged, serrated ovipositor [14]. However, in contrast to *D. suzukii*'s ripe fruit preference, *D. subpulchrella* shows an intermediate preference for ovipositing in ripening fruits [15,16], suggesting that strict ripe fruit preference is a recently evolved adaptive novel trait. *D. subpulchrella* and *D. suzukii* diverged approximately 2–3 million years ago (Fig 1A), whereas their divergence from *D. melanogaster* occurred approximately 13 million years ago [17]. This divergence, coupled with the shift in oviposition preferences, establishes a stepwise evolutionary system [15,16] that can be used to address mechanisms of adaptation. Because *D. melanogaster* has only diverged 13 million years ago*,* its genetic toolkit can still be leveraged within this system [18]. Therefore,

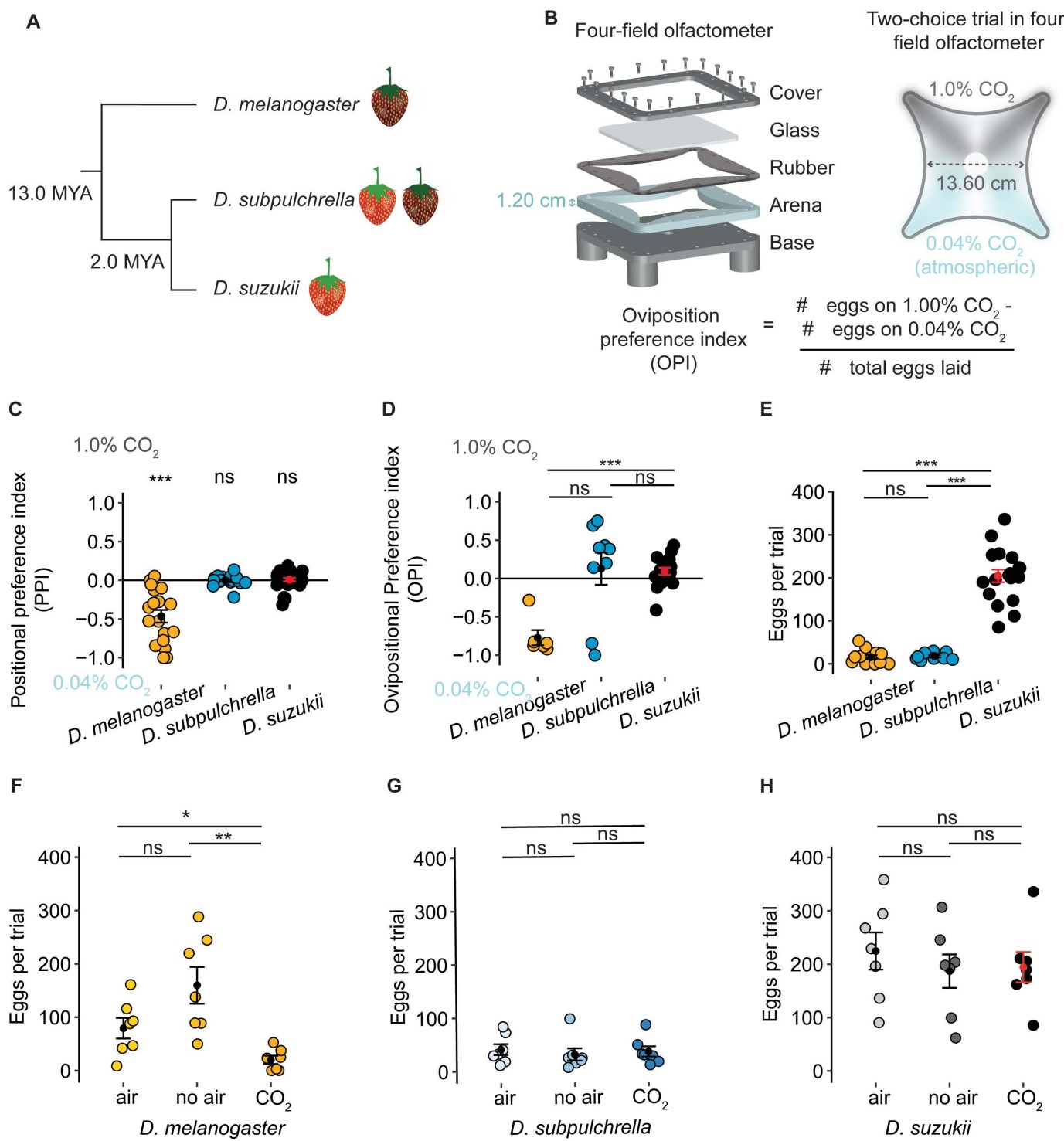

**Fig 1. Behavioral responses to 1% $CO_2$ between *D. suzukii*, *D. subpulchrella*, and *D. melanogaster*.** A) Species tree of the three focal species. *D. subpulchrella* and *D. suzukii* are closely related and have only diverged from each other 2 million years ago (MYA). B) Schematic design of the four-field olfactometer. In our design we increased the arena size to 1.2 cm to accommodate an egg-laying substrate during experiments and incorporated a rubber layer for a secure seal. This chamber was used to perform two-choice egg-laying trials between 0.04% $CO_2$ and 1% $CO_2$. C) Positional preferences of adult mated flies in the four-field olfactometer for the three species (Dmel = 18 trials, Dsub = 25 trials, Dsuz = 27 trials). *D. melanogaster*

preferred to spend time on the 0.04% $CO_2$ side (Wilcoxon test with Holm correction, Dmel $Q < 0.001$, Dsub $Q = 0.65$, and Dsuz $Q = 0.32$). D) Oviposition preference index (OPI) for 1% $CO_2$ of the three species in the four-field olfactometer (Dmel = 6 trials, Dsub = 9 trials, Dsuz = 18 trials) with 12 female flies per experiment. *D. suzukii* and *D. subpulchrella* were less aversive to 1% $CO_2$. *D. suzukii* and *D. subpulchrella* had a significantly different behavior from each other (Kruskal-Wallis test followed by pairwise Wilcoxon signed-rank test with Holm correction, Dmel-Dsuz: $Q = 8.9e-05$, Dmel-Dsub: $Q = 0.072$). E) The total eggs counted in each trial conducted in C (Dmel = 12 trials, Dsub = 9 trials, Dsuz = 18 trials). For *D. melanogaster,* there were many trials that ended in zero eggs, making it impossible to calculate an OPI; these trials were included in this panel. *D. suzukii* lay significantly more eggs than *D. subpulchrella* and *D. melanogaster* (Kruskal-Wallis test followed by pairwise Wilcoxon signed-rank test with Holm correction, $Q = 1.4e-05$ and $Q = 6.2e-05$, respectively). F, G, H) Three simultaneous trials were conducted with flies collected on the same day from the same vials for each species. Flies were placed in either air, no air, or the original $CO_2$ trials. *D. melanogaster* laid significantly fewer eggs in the 1% $CO_2$ trials compared to the air (Kruskal-Wallis test followed by pairwise Wilcoxon signed-rank test with Holm correction, $Q = 0.036$) or no air trials ($Q = 0.008$). The other species did not lay fewer eggs in 1% $CO_2$. ***$Q \leq 0.001$; **$Q \leq 0.01$; *$Q \leq 0.05$; ns, $Q > 0.05$; and this notation is used throughout all figures.

the *suzukii* species complex offers a tractable system for investigating the genetic and neural changes underlying a novel egg-laying behavior that facilitates the exploitation of a novel ecological niche.

Multiple evolutionary mechanisms have been proposed as plausible foundations for adaptation. For example, non-synonymous mutations in coding regions can have large phenotypic effects [19,20], and even a single amino acid change in an odorant receptor can shift ligand sensitivity between *Drosophila* species [21]. Yet changes in protein-coding regions can also have negative consequences through pleiotropy [22]. By contrast, modifications in gene regulation [23–29] can circumvent these costs and may provide a less risky evolutionary path [28]. Indeed, adaptive mutations in regulatory regions have been repeatedly demonstrated, particularly in the context of morphological evolution. Differences in traits such as wing shape, larval morphology, and pigmentation between *Drosophila* species have been traced to *cis*-regulatory elements [26,27,30]. The conditions under which evolution favors coding versus regulatory changes, however, remain unclear.

Ripening and rotting fruits present distinct chemical and textural profiles. Ripening fruits are harder, less sweet, and contain lower concentrations of ethanol and acetic acid [16,31,32]. To interact with these chemical profiles, flies use precisely tuned chemosensory receptor genes at the peripheral sensory system [33]. Studies have linked adaptive behavioral shifts with changes in both the *Drosophila* peripheral sensory system and central nervous system [15,16,21,31,32,34–37]. To identify candidate sensory genes that may explain *D. suzukii*'s adaptive shift toward ovipositing in ripening fruits, we previously identified genes under selection and significantly differentially expressed between *D. suzukii*, *D. subpulchrella*, and *D. melanogaster* [16]. One gene of particular interest to us was the $CO_2$ receptor *Gr63a*, as it was under positive selection in *D. suzukii* and was significantly highly expressed in *D. subpulchrella,* suggesting that two distinct genetic mechanisms have evolved between the sister species. Another reason this gene was of interest is that ripening fruits release more $CO_2$ as a by-product of respiration [38,39]. We hypothesize that *D. suzukii* and *D. subpulchrella* have evolved separate evolutionary mechanisms—one involving changes in protein-coding regions and the other involving regulatory changes—to adapt to the $CO_2$-rich environment of ripening fruits.

Drosophilids detect $CO_2$ through the co-expression of the seven-transmembrane gustatory receptors *Gr63a* and *Gr21a* [40,41]. These receptors are located on the third antennal segment. *Gr63a* and *Gr21a* are expressed on the ab1C neuron, which is housed in the ab1 sensillum, where the heterodimer is consistently tuned to detect $CO_2$ [42]. Because the ab1C neuron of *D. suzukii* exhibits heightened $CO_2$ sensitivity [43], this system provides a powerful framework for examining how evolutionary changes in regulatory or coding sequences contribute to divergence in sensory receptor function. In this study, we investigated both behavioral responses to $CO_2$ and the genetic mechanisms underlying these derived phenotypes.

## Results

### *D. suzukii* and *D. subpulchrella* do not show aversion to ovipositing in 1% $CO_2$

*D. melanogaster* naturally exhibits aversive behavior to $CO_2$ in short-duration preference assays [39,43], yet the influence of $CO_2$ on oviposition behaviors has not been studied in *D. melanogaster* or the *D. suzukii* species complex. To study

whether these species (Fig 1A) exhibit similar aversive responses to $CO_2$, we constructed a four-field olfactometer [44], with an elevated arena height (1.2 cm instead of 0.7 cm) to accommodate an egg-laying substrate (Fig 1B). This experimental setup was designed to mimic a more natural environment where flies can move freely while exposed to $CO_2$ variations that represent early and late ripening stages [38,39]. In the four-field olfactometer, flies were first given the choice between atmospheric $CO_2$ level (0.04%) and 1% $CO_2$, chosen as a high-end value for ripe fruit, ripe fruit naturally reaches only around 0.16% $CO_2$ levels at the ripe stage [38]. As proof of concept that the chamber creates a high and low $CO_2$ environment, we first recorded the positional preference of flies between high and low $CO_2$ concentrations using 15-minute assays (Fig 1C). We found that *D. melanogaster* flies tended to avoid the 1% $CO_2$ side, consistent with previous reports [39]. Conversely, both *D. subpulchrella* and *D. suzukii* showed no aversion to 1% $CO_2$ (mean PPI of -0.61, -0.11, and 0.04, respectively; *Q-values* were < 0.001, 0.65, and 0.32, respectively; Fig 1C), suggesting a behavioral shift in $CO_2$ response.

Next, we examined whether oviposition preferences differed between the ripe fruit-loving species and the overripe fruit-loving species *D. melanogaster*. The base of the chamber was covered with an agarose layer, and flies were given a choice to oviposit at atmospheric $CO_2$ levels or 1% $CO_2$ over a 16-hour period. *D. melanogaster* showed a strong aversion, with a negative preference index indicating its preference for the low-$CO_2$ environment (mean OPI = -0.77 ± 0.10 SEM [standard error of the mean]), whereas *D. suzukii* and *D. subpulchrella* did not (mean OPIs ± (SEM) are 0.10 ± 0.05 and 0.13 ± 0.21, respectively; Fig 1D). The oviposition preference of *D. melanogaster* was significantly different from *D. suzukii* (*Q-value* = 8.9e-05), while *D. subpulchrella* did not differ significantly from either species (*Q-value* = 0.07; *Q-value* = 0.18). Unlike *D. melanogaster*, *D. subpulchrella* did not show a strong aversion to elevated $CO_2$; instead, its responses were more variable, possibly reflecting its intermediate preference for ripe fruit. Overall, *D. melanogaster* exhibited consistent avoidance of 1% $CO_2$, whereas both *D. suzukii* and *D. subpulchrella* were equally likely to oviposit on either side. Both *D. melanogaster* and *D. subpulchrella* had significantly lower egg-laying rates in the overnight trials compared to *D. suzukii* (Kruskal-Wallis test followed by pairwise Wilcoxon signed-rank test with Holm correction, *Q-value* = 1.4e-05 and *Q-value* = 6.2e-05, respectively; *Q-values* are adjusted p-values; Fig 1D). These results suggest that *D. suzukii* has a novel egg-laying behavior that is not suppressed by high $CO_2$ concentrations (Fig 1E).

To rule out air flow as a confounding factor and confirm that $CO_2$ drives species-specific differences in egg-laying avoidance or suppression, we tested whether air flow influences oviposition in *D. subpulchrella*, *D. suzukii*, and *D. melanogaster*. To address this, we conducted three simultaneous trials. In these trials, flies from the three species were exposed to either atmospheric $CO_2$ ("air" condition) from all corners, no additional air flow ("no air" condition), or the same two-choice $CO_2$ described earlier ("$CO_2$" condition). *D. melanogaster* laid significantly fewer eggs in the 1% $CO_2$ trials compared to the air condition (Kruskal-Wallis test followed by pairwise Wilcoxon signed-rank test with Holm correction, *Q-value* = 0.04) or the no air condition (*Q- value* = 0.01), confirming that $CO_2$ suppresses egg laying in this species, not air flow (Fig 1F). *D. subpulchrella* laid significantly fewer eggs than *D. suzukii* but deposited similar numbers across all three conditions, indicating that while it may have lower fecundity, it does not exhibit aversion to 1% $CO_2$ (Fig 1G). *D. suzukii* also laid consistent numbers of eggs across all three conditions (Fig 1H), confirming that air is not a confounding factor in this species. These results suggest that both *D. subpulchrella* and *D. suzukii* have evolved a new aspect of egg-laying behavior; these species do not avoid $CO_2$ and do not show $CO_2$-induced suppression of egg-laying.

### *D. subpulchrella* and *D. suzukii*'s ab1C neuron is more sensitive to $CO_2$

After confirming that *D. suzukii* and *D. subpulchrella* exhibit greater tolerance for ovipositing at elevated $CO_2$ levels, we investigated whether this shift results from changes in neural sensitivity. The two $CO_2$ co-receptors, *Gr63a* and *Gr21a* are expressed on the ab1C neuron, which is housed in the ab1 sensilla. As sensilla are readily accessible in *Drosophila,* we can directly record neural activity of ab1C neurons using single sensillum electrophysiology (SSE). To compare wildtype ab1C responses to $CO_2$ across the three species, we recorded from ten ab1 sensilla for each species. We analyzed the

spikes before, during, and after $CO_2$ delivery (Fig 2A, 2C, and 2E) and recorded the local field potentials of each recording (Fig 2B, 2D, and 2F). The average spike rate across 100 ms after $CO_2$ onset and the maximum spike rate at $CO_2$ onset—which represents the spike rate directly after $CO_2$ delivery and the neurons sensitivity to the odor—were higher in both *D. suzukii* (Figs 2G, S1A and S1B; Kruskal-Wallis test followed by pairwise Wilcoxon signed-rank test with Holm correction, *Q-value* = 0.001) and *D. subpulchrella* (*Q-value* = 0.01) compared to *D. melanogaster* (Fig 2G). These results indicate that ab1C neurons of these species are more responsive to high $CO_2$ concentrations. *D. subpulchrella*, however, exhibited a larger range, like its oviposition behavior.

The local field potential (LFP), which refers to the collective electrical activity of the recorded neuron as well as nearby neurons, was also recorded. While the signal in the antenna is not well isolated due to the high density of closely packed sensilla [45], there is evidence that the slope of the downward deflection of the LFP correlates with receptor potential during odor stimulation [45]. For the LFP, the slope at $CO_2$ onset was also lower in *D. suzukii* (Kruskal-Wallis test followed by pairwise Wilcoxon signed-rank test with Holm correction, *Q-value* = 0.001) and *D. subpulchrella* (*Q-value* = 0.01) compared to *D. melanogaster* (Fig 2H), further supporting that *D. suzukii* and *D. subpulchrella* are more sensitive to $CO_2$.

The baseline spike rate is comparable across species (S1C and S1D Fig). Similarly, the steady-state spike rate, which reflects a balance between factors promoting channel opening and those promoting channel closure, remains the same across species (S1E and S1F Fig), suggesting that the channel structure has not changed significantly. Taken together, these results demonstrate that the ab1C neuron is more sensitive to $CO_2$ in *D. suzukii* and *D. subpulchrella* than in *D. melanogaster*.

**Transgenic *D. melanogaster* flies expressing *D. subpulchrella* and *D. suzukii* Gr63a CDS or *cis*-regulatory element**

Our previous work [16] suggested that changes in the coding sequence or *cis*-regulatory element of the *Gr63a* gene could both influence the flies' peripheral sensory system. To confirm an increase in *Gr63a* expression in *D. subpulchrella*, we conducted a qPCR analysis of the *Gr63a* expression levels in antennal tissues of the three species. Using the TATA-binding protein (TBP) as an endogenous control due to its consistent expression patterns across the three species (S1 Table), we detected a significant difference in *D. subpulchrella Gr63a* expression compared to *D. suzukii* and *D. melanogaster*. Specifically, the relative quantification (RQ) revealed a five-fold increase in *D. subpulchrella* (mean RQ = 5) compared to *D. suzukii* (mean RQ = 1.2) and *D. melanogaster* (mean RQ = 1) (Fig 3A). These qPCR findings corroborate the role of the *D. subpulchrella*'s *cis*-regulatory element in heightening *Gr63a* expression.

We also compared the *Gr63a* coding sequence between *D. suzukii*, *D. subpulchrella*, and *D. suzukii*, by aligning the sequences (S3A Fig) and found that a substantial number of substitutions in the amino acid sequence have evolved at the C-terminus (Figs 3B and S3A). Insect odorant receptors have an extracellular c-terminal that may be involved in heterodimer formation and therefore influence receptor complex integrity [46]. While the *D. suzukii* sequence differs substantially (95.3% identity) from that of *D. melanogaster*, *D. subpulchrella* and *D. suzukii* only differ from each other at three amino acid sites: sites 48, 61, and 440 (S3A Fig). Sites 48 and 61 are in the intracellular N-terminal region. Single amino acid substitutions in the N-terminus of insect ORs have previously been shown to alter receptor function, likely through effects on folding, trafficking, or assembly [47]. While it is unclear exactly what role this region may play in receptor function, its intracellular nature could contribute to downstream trafficking or receptor formation.

To test these changes in CDS and gene regulation, we constructed plasmids containing the GAL4 element fused with each of the species-specific *cis*-regulatory elements and plasmids containing the UAS element with each species' *Gr63a* CDS in the *D. melanogaster* Gr63a null background. A similar process was previously utilized in mosquito $CO_2$ receptor evolution studies [48]. The GAL4 constructs were inserted on the 3R chromosome, while the UAS constructs were inserted on the second chromosome. Gr63a is located on chromosome 3L. To place the three *cis*-regulatory GAL4 lines on the same chromosome as the Gr63a[1] line background, which carries a non-functional $CO_2$ receptor due to a null mutant allele of Gr63a [41], we used balancers to create recombination events so that both elements could exist in one

 

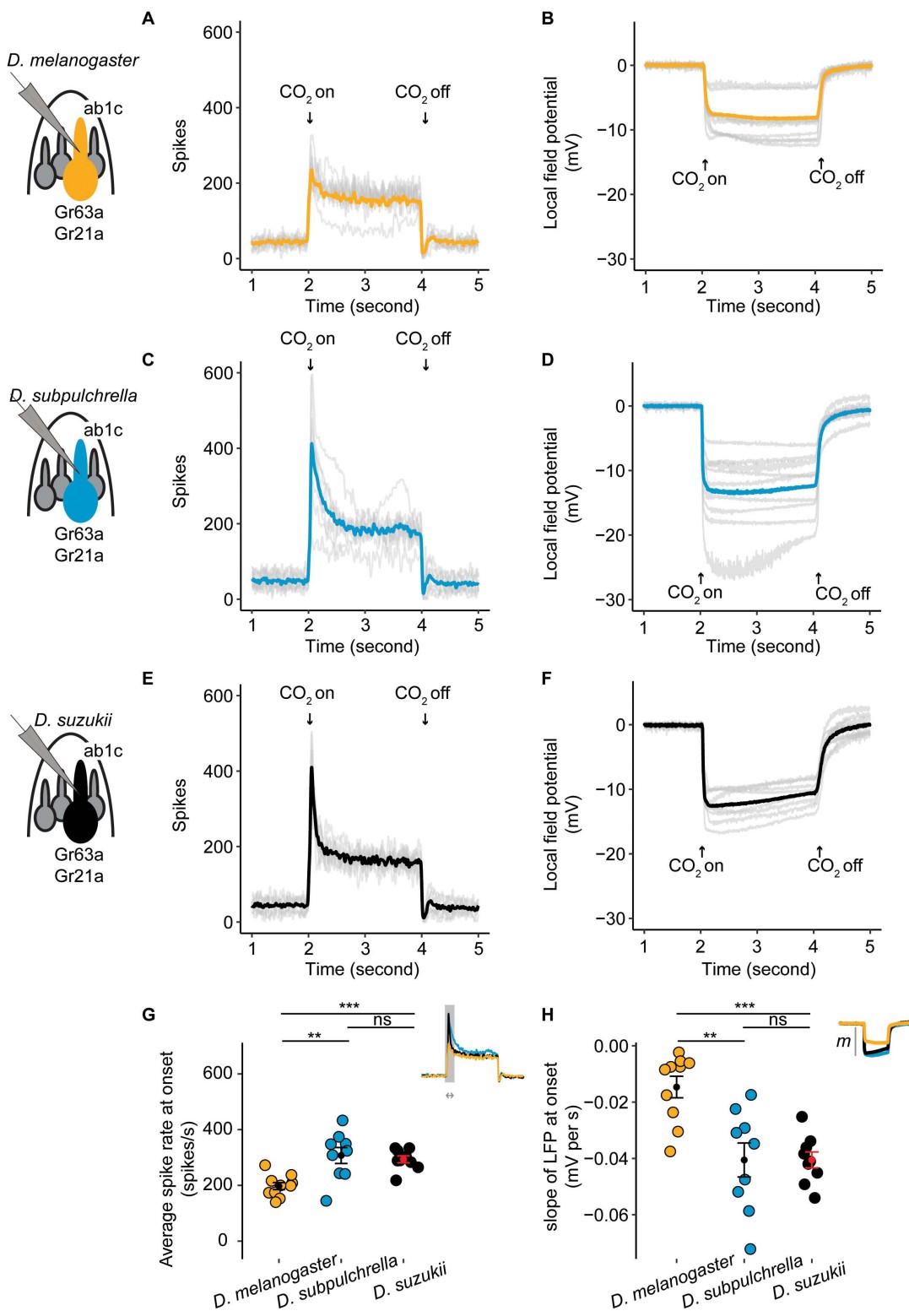

**Fig 2. Single sensillum electrophysiology recordings for the ab1C neuron in wildtype *D. suzukii*, *D. subpulchrella*, and *D. melanogaster* flies.** A,C,E) *D. melanogaster*, *D. subpulchrella* and *D. suzukii* peristimulus time histograms. The ab1C spike rate is recorded on the y-axis and time in milliseconds (ms) is recorded on the x-axis. The $CO_2$ delivery was delivered for 2 seconds at a time. Each grey line is the average of five replicates of

one ab1C neuron recording. The yellow, blue, and black lines are the average across the ten replicates. B,D,F) *D. melanogaster*, *D. subpulchrella* and *D. suzukii* LFP recordings for each ab1C recording in A,C,E. Each grey line is the average of five replicates from one ab1C neuron and the yellow, blue, and black lines are the average across ten neurons. G) The average ab1C spike rate across 100 ms at the $CO_2$ onset for each species. *D. suzukii* and *D. subpulchrella* had a significantly higher spike rate than *D. melanogaster* (Kruskal-Wallis test followed by pairwise Wilcoxon signed-rank test with Holm correction, $Q = 0.001$ and $Q = 0.012$). H) The slope of the LFP at $CO_2$ onset calculated across 320 ms. *D. subpulchrella* and *D. suzukii* had a faster rate of change than *D. melanogaster* (Kruskal-Wallis test followed by pairwise Wilcoxon signed-rank test with Holm correction, $Q = 0.001$ and $Q = 0.008$).

fly. We then crossed these flies with the three possible UAS-CDS lines, resulting in a total of nine combinations (Fig 3C). Using these combinations, we tested the effects of different CDS and *cis*-regulatory elements on ab1C spike rate and $CO_2$ oviposition preference within a shared transgenic background.

## ab1C responses of the transgenic lines

We conducted de-identified SSE experiments on the nine transgenic fly lines carrying GAL4 and UAS combinations in *Gr63a* null background [41]. We used flies that were homozygous for both the GAL4 and UAS transgenes, except for one line expressing the *D. suzukii* Gr63a CDS with the *D. subpulchrella* Gr63a *cis*-regulatory element. For this line, we recorded from flies that were homozygous for the GAL4 and heterozygous for the UAS in the $Gr63a^1$ background, due to lethality.

The ab1C neurons in all nine transgenic lines responded to 1% $CO_2$ (Fig 4A). Spike rates at $CO_2$ onset were extracted for all replicates, and mean values with standard errors were calculated for each line. The statistical groupings above the bars summarize which lines differed significantly after pairwise comparisons. Spike rates at $CO_2$ onset varied significantly across lines (Fig 4B). Two groups differed most clearly: flies expressing the *D. suzukii* coding sequence showed slightly elevated spike rates, whereas flies expressing the *D. subpulchrella* coding sequence showed lower spike rates.

To identify the elements driving differences in $CO_2$ sensitivity, we summarized the spike rates at $CO_2$ onset across all combinations of *cis*-regulatory elements and coding sequences using a heatmap (Fig 4C). When comparing the CDS between species, *D. suzukii* differed significantly from both *D. melanogaster* and *D. subpulchrella* (*Q-value* = 0.001 vs MEL; *Q-value* = 0.011 vs SUB) (Kruskal–Wallis test followed by pairwise Wilcoxon rank-sum tests with Holm correction), whereas *D. melanogaster* and *D. subpulchrella* did not differ (*Q-value* = 0.50). The addition of the *D. suzukii* CDS in a transgenic fly consistently increased the spike rate. Although the *D. subpulchrella cis*-regulatory element trended toward higher spike rates, this increase was not statistically significant.

As we recorded from heterozygous individuals for the DsubGr63a^cis^-GAL4 > UAS-DsuzGr63a^cds^ line, it is possible that the signal was confounded, as these flies carried the *D. suzukii* coding sequence on only one of their two chromosomes. To correct for the heterozygous recording, we used a model to estimate the homozygous spike rate for that cross (see Methods). To validate the model, we applied the same approach to a line with known homozygous spike rate values, DsubGr63a^cis^-GAL4 > UAS-DsubGr63a^cds^ (S3B Fig). The predicted and recorded values performed similarly (Figs 4B and S3B) (Kruskal-Wallis test followed by pairwise Wilcoxon signed-rank test with Holm correction, *Q-value* = 0.19; mean of B = 223.721; mean of estimated B = 228.3726), confirming the robustness of our model (Fig 4B).

After incorporating the modeled value for DsubGr63a^cis^-GAL4 > UAS-DsuzGr63a^cds^ (S3C Fig), and comparing spike rates across all homozygous individuals, we identified two elements that significantly increased $CO_2$ sensitivity. The strongest increase was caused by the *D. suzukii* Gr63a CDS (Kruskal-Wallis test followed by pairwise Wilcoxon signed-rank test with Holm correction, Dmel–Dsuz: *Q-value* = 1.1e-08 and Dsub-Dsuz: *Q-value* = 6.3e-07). The *D. subpulchrella* Gr63a *cis*-regulatory element also increased sensitivity, though more modestly (Kruskal-Wallis test followed by pairwise Wilcoxon signed-rank test with Holm correction, *Q-value* = 0.01 between Dmel and Dsub and *Q-value* = 0.004 between Dsub and Dsuz; S3C Fig). When restricting the analysis to rows with only homozygous recorded values and removing any modeled values (S3D and S3E Fig), the pattern remained consistent: both the *D. subpulchrella cis*-regulatory element

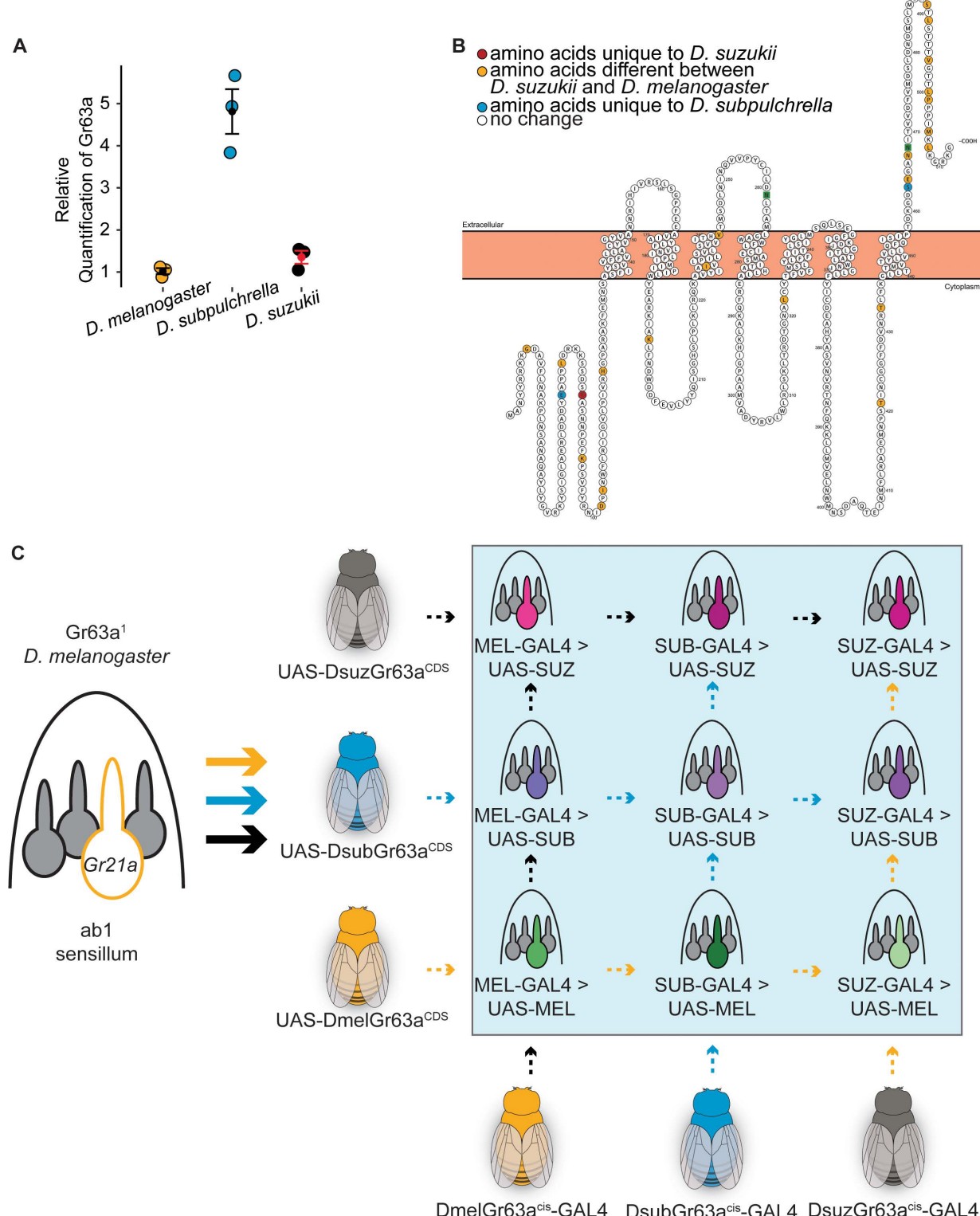

**Fig 3. Expression and structural properties of Gr63a and schematic representation of transgenic line generation.** A) qPCR results targeting *Gr63a* with antennal RNA from the three species. The relative quantification (RQ) revealed a five-fold increase in *D. subpulchrella* (mean = 5) compared to *D. suzukii* (mean = 1.2) and D. melanogaster (mean = 1). B) The *D. melanogaster* Gr63a protein with the amino acids that differ between *D.*

*melanogaster* and *D. suzukii's* are highlighted in yellow, in red are the amino acids that are unique to *D. suzukii*, and in blue are the amino acids that are unique to *D. subpulchrella*. Most of the changes have occurred at the C-terminal. C) Schematic of transgenic line generation. GAL4 driver lines carrying each species' *cis*-regulatory element were crossed with a *Gr63a¹* knockout line and then with one of three UAS lines containing the Gr63a coding sequence (CDS) from each species cloned into a UAS vector. This generated nine transgenic lines in total (highlighted in the blue box), representing all combinations of *Gr63a cis*-regulatory elements driving each species' *Gr63a* CDS. The names in the blue box are the abbreviations used in later figures.

(Kruskal-Wallis test followed by pairwise Wilcoxon signed-rank test with Holm correction, Dsub-Dmel: *Q-value* = 0.01 and Dsub-Dsuz: *Q-value* = 0.0041) and the *D. suzukii CDS* (Kruskal-Wallis test followed by pairwise Wilcoxon signed-rank test with Holm correction, Dsuz-Dsub: *Q-value* = 6.3e-07 and Dsuz-Dmel: *Q-value* = 1.1e-08) significantly increased the spike rate.

Flies expressing the *D. suzukii* CDS under the control of the *D. subpulchrella cis*-regulatory element in *Gr63a¹* background also showed the fastest rate of change in LFP recordings (S2B and S2C Fig). In addition, flies expressing the *D. melanogaster* CDS had a faster rate of change than those expressing the *D. subpulchrella* CDS. This difference was not supported by the spike rate data and likely reflects the contribution of nearby neurons to the LFP signal. Because the LFP represents the summed extracellular currents from all neurons within a sensillum rather than the membrane potential of a single neuron [45], variations in neighboring neuronal activity can influence the LFP independently of the $CO_2$-sensitive neuron's firing. Therefore, LFP and spike rate measurements are not always expected to align.

To evaluate whether differences in SSE responses were associated with transgene expression levels, we quantified Gr63a transcript levels in the nine transgenic lines used for electrophysiological recordings. Quantitative RT-PCR revealed variation in Gr63a expression among lines, consistent with predictions (S4A Fig). Across six lines for which we were able to obtain a consistent number of homozygous flies needed for the Quantitative RT-PCR, Gr63a expression levels showed a positive correlation with the magnitude of SSE responses (Pearson's R = 0.62, S4B Fig), indicating that higher transcript abundance was generally associated with stronger neural activity in *D. melanogaster*. These findings suggest that variation in transgene expression, likely driven by differences in enhancer strength or genomic context, contributes to the observed variability in SSE responses.

Overall, these results highlight the critical roles of both coding sequences and *cis*-regulatory elements in modulating $CO_2$ sensitivity in the Gr63a neurons of *D. subpulchrella* and *D. suzukii*. The *D. suzukii* CDS had the strongest effect, whereas the *D. subpulchrella cis*-regulatory element produced a more modest but detectable increase in spike rate. Because the *D. subpulchrella* coding sequence did not yield a similar increase, the modest elevation associated with the *cis*-regulatory element, also supported by our modeling, may underlie the increased $CO_2$ sensitivity observed in *D. subpulchrella*. Together, these findings indicate that coding sequence mutations exert the strongest functional effects on receptor activity.

### Two-choice egg-laying trials with transgenic lines

After determining that the *D. suzukii* CDS and *D. subpulchrella cis*-regulatory element were sufficient to increase ab1C sensitivity, we tested the oviposition preference of the transgenic lines for 1% $CO_2$ in the four-field olfactometer. Heterozygous individuals were used for four transgenic lines that had low or no homozygous hatching rates (Fig 5A and 5B).

The *Gr63a¹* knockout strain, which lacks $CO_2$ sensitivity, showed no oviposition preference between the low and high $CO_2$ environments. In contrast, the patterns in the other transgenic lines were less clear, though the lines generally showed greater aversion to 1% $CO_2$ (Fig 5A). Interpretation of oviposition preference was confounded by a significant reduction in egg laying per trial (Fig 5B), which limited our ability to detect reliable trends in the oviposition preference index (Fig 5A). This reduction in egg laying is particularly notable because the knockout strain, which exhibits no aversion to 1% $CO_2$ did not show such a decrease. Transgenic flies exhibited $CO_2$ aversion similar to, or greater than, that of

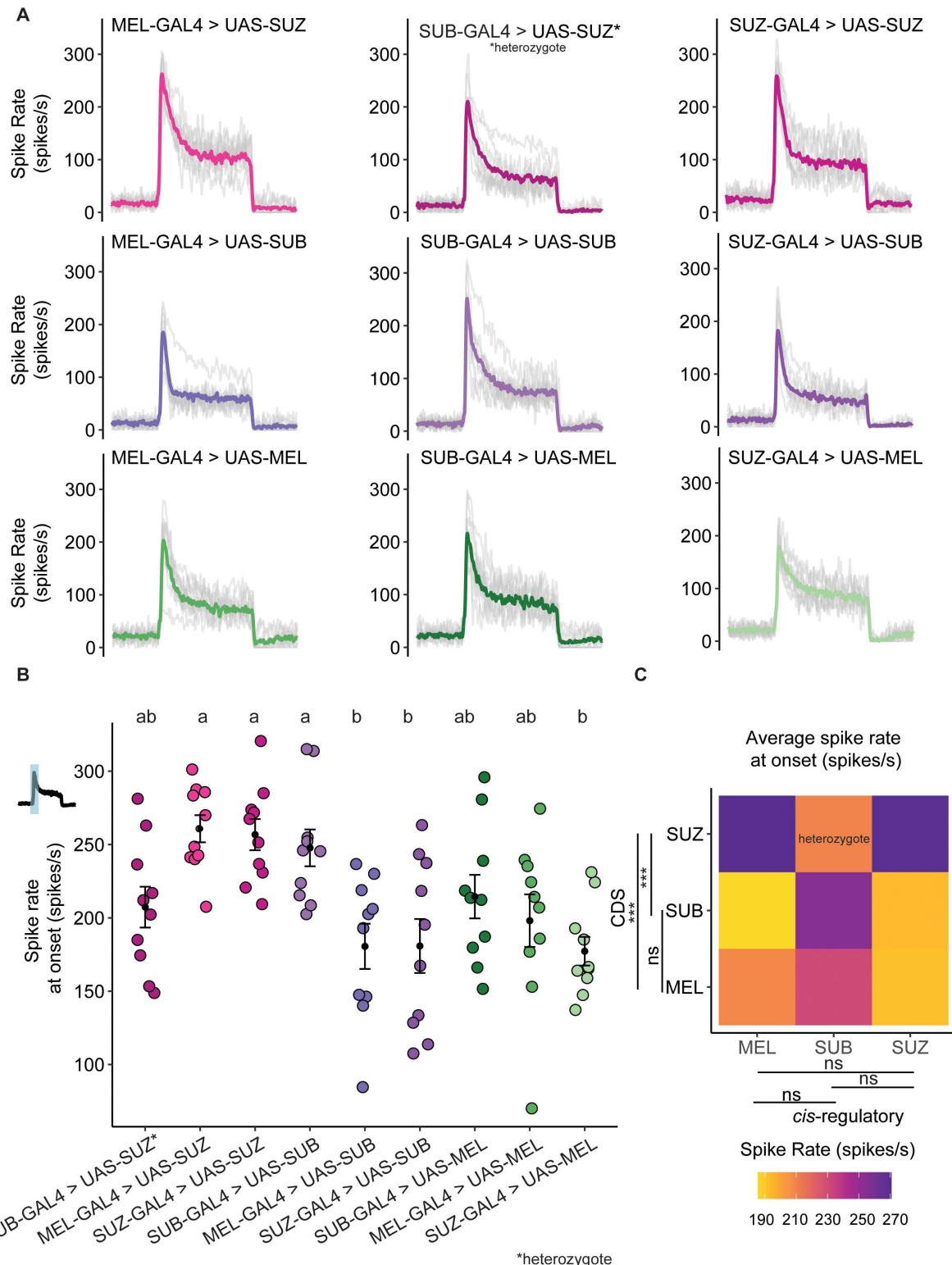

**Fig 4. ab1C neuron spike rate in cross-species sensory substitution lines.** A) The peristimulus time histograms of each transgenic lines (n=10 ab1C neuron with 5 recordings for each neuron). One transgenic line, SUB-GAL4 > UAS-SUZ, is homozygous lethal, for this line we recorded from heterozygous flies; These flies still have the balancer phenotype of *CyO*. B) The average spike rate at $CO_2$ onset (across 20 ms). Letters above panel B

denote statistical groups determined by one-way ANOVA followed by 36 pairwise Tukey comparisons among the nine lines. Lines sharing at least one letter (e.g., 'ab' vs. 'a' or ab' vs. 'b') are not significantly different; lines without shared letters (e.g., 'a' vs. 'b') differ significantly. C) Heatmap of average spike rate at $CO_2$ onset across all combinations of GAL4 drivers and UAS coding sequences. CDS comparisons revealed that *D. suzukii* differed significantly from both *D. melanogaster* and *D. subpulchrella* (Q = 0.001 vs MEL; Q = 0.011 vs SUB; Kruskal–Wallis test followed by pairwise Wilcoxon rank-sum tests with Holm correction), whereas *D. melanogaster* and *D. subpulchrella* did not differ (Q = 0.50).

wildtype *D. melanogaster*. These observations suggest that an additional behavioral switch, acting downstream of the peripheral sensory system, modulates $CO_2$-dependent oviposition preference.

Lines carrying the *D. suzukii* CDS showed the strongest decrease in egg laying, which differed significantly from lines expressing the *D. melanogaster* CDS (*Q-value* = 0.04, Fig 5B–5C). These findings suggest that these flies may have the highest $CO_2$ sensitivity, consistent with the SSE results. All three lines carrying the *D. subpulchrella cis*-regulatory element failed to hatch or produced few homozygous adults, making it difficult to perform experiments with more than 10 flies per trial; for these lines, heterozygous individuals were also used.

### *D. subpulchrella*, *D. suzukii*, and *D. melanogaster Gr63a* are all expressed in the V-glomerulus

To determine whether Gr63a-expressing sensory neurons exhibit similar brain expression patterns across species, we used our species-specific Gr63a-GAL4 lines to express UAS-GFP and imaged the brains. We found that Gr63a[suzukii]-GAL4, Gr63a[subpulchrella]-GAL4, and Gr63a[melanogaster]-GAL4 all labeled gustatory neurons in the V-Glomerulus (Fig 5D). These findings suggest that the observed behavioral differences in $CO_2$-driven oviposition preference may be regulated downstream of the antennal lobe projection neurons or involve specific interneurons innervating the antennal lobe. Understanding the neural basis of this $CO_2$ preference switch presents an intriguing future direction. However, addressing this question would require calcium imaging in the endogenous fly species, which may be challenging in the non-model *Drosophila* species, *D. subpulchrella* and *D. suzukii*.

## Discussion

In summary, we determined how changes in gene sequence and expression affect the peripheral sensory receptors that mediate $CO_2$ sensitivity and oviposition behavior in the *Drosophila suzukii* species complex (Fig 5E). Unlike other odorant receptors, which have evolved coding sequence (CDS) modifications that shift ligand receptivity [21], $CO_2$ receptors in *Drosophila* have remained consistently sensitive across species [42]. This system therefore provided a unique opportunity to assess the relative contributions of *cis*-regulatory and CDS changes to sensory adaptation. Our results demonstrate that *D. suzukii* and *D. subpulchrella* exhibit decreased aversion yet increased sensitivity to $CO_2$, a pattern that initially seemed counterintuitive, as we expected lower $CO_2$ sensitivity to correlate with reduced avoidance. Given that ripe fruit emits higher $CO_2$ levels than rotting fruit [38], *D. suzukii* and *D. subpulchrella* may rely on $CO_2$ as an ecological cue to locate suitable oviposition sites. $CO_2$ also plays a role in physiological signaling, such as stress responses in *D. melanogaster*, where stressed flies release $CO_2$ as part of an aversive odorant plume [49]. The reduced $CO_2$ aversion observed in *D. suzukii* and *D. subpulchrella* may thus reflect a broader shift in how these species interpret $CO_2$ as an environmental signal.

Electrophysiological recordings revealed that *D. suzukii* and *D. subpulchrella* have heightened $CO_2$ sensitivity compared to *D. melanogaster* (Fig 5E). Using transgenic *D. melanogaster* lines, we found that species-specific modifications in the *Gr63a* coding sequence of *D. suzukii* and *cis*-regulatory changes of *D. subpulchrella* were sufficient to increase $CO_2$ sensitivity. These findings highlight that two closely related species have used different genetic mechanisms to achieve similar increases in spike rate. Our results also support that changes in gene regulatory elements may produce more subtle phenotypes compared to protein-coding sequence changes. While changes in the CDS may be essential for the specialist species (*D. suzukii*), it could be more advantageous for the intermediate species (*D. subpulchrella)* to evolve

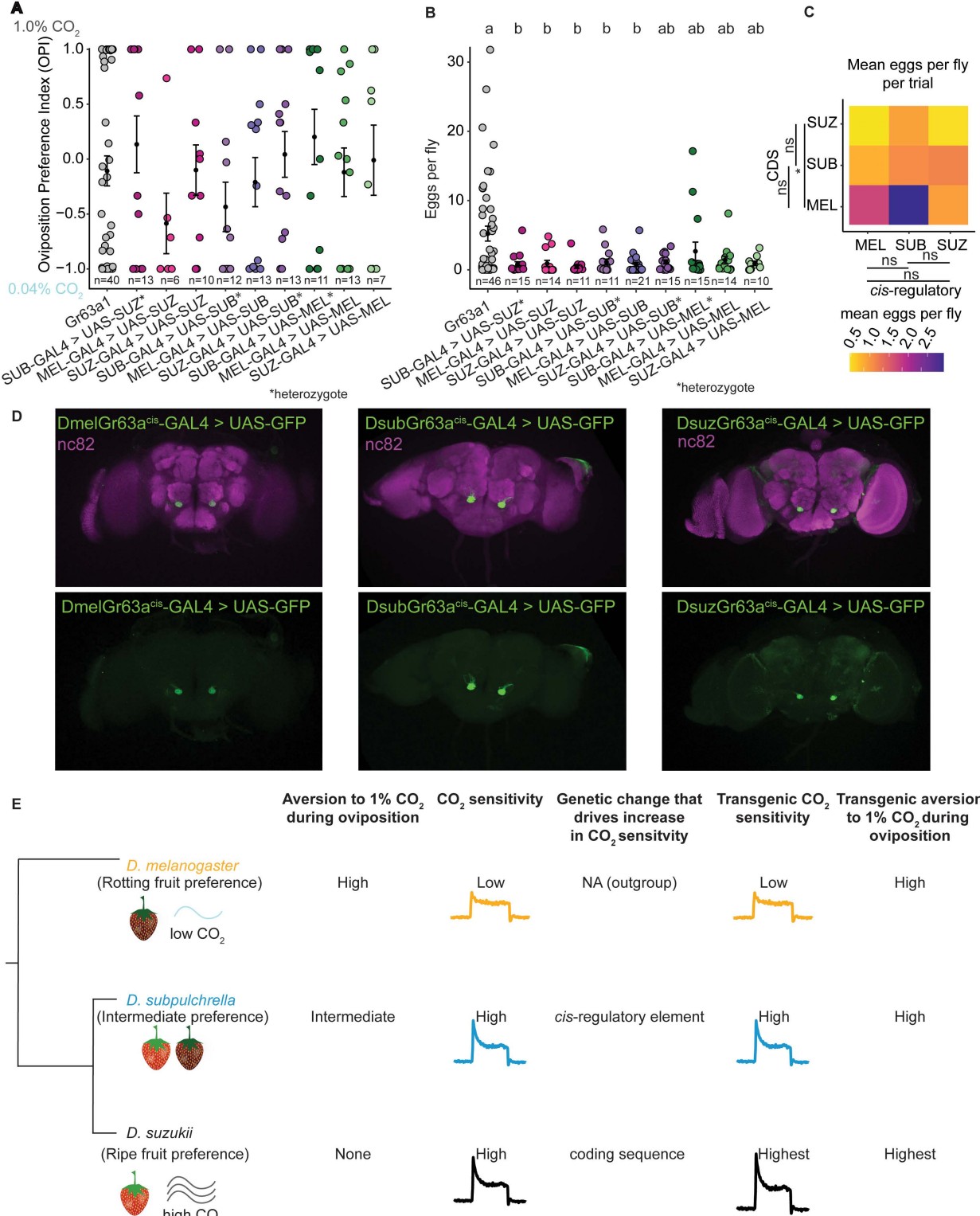

**Fig 5. Convergent behavior driven by divergent genetic processes.** A) Two-choice oviposition preference trials with 0.04% $CO_2$ and 1.0% $CO_2$ for transgenic lines. The oviposition preference index was quantified for cross-species sensory substitution lines. A preference of –1 indicates egg laying on 0.04% $CO_2$, while a preference of 1 indicates egg laying on 1.0% $CO_2$. Flies from the *Gr63a* knockout line (grey) showed no preference, while transgenic

lines exhibited variable responses. B) Number of eggs laid in each two-choice trial from (A). Only the knockout line laid substantial numbers of eggs. Different letters indicate groups that differ significantly based on Tukey's HSD test (p < 0.05). Groups that share a letter are not significantly different. All lines laid fewer eggs than the *Gr63a* knockout strain. C) Heat map comparing the mean egg-laying rate by GAL4 and UAS element expressed. When comparing the GAL4 and UAS elements expressed in each fly line, those carrying the *D. suzukii* CDS showed the strongest decrease (Kruskal–Wallis test followed by pairwise Wilcoxon signed-rank test with Holm correction, Dsuz–Dsub: Q = 0.27; Dsuz–Dmel: Q = 0.04). D) All three species' *cis*-regulatory elements drive GFP expression in the V-glomerulus. E) Summary model showing how either the *D. suzukii* CDS or *D. subpulchrella cis*-regulatory element alters the ab1C response and each species' oviposition behavior in the context of $CO_2$.

*cis*-regulatory modifications, allowing it to adjust sensitivity as needed. Changes in the CDS and regulatory sequences can have additive effects on function, as flies with both elements likely exhibit the highest $CO_2$ sensitivity.

These differences in sensitivity may provide insight into why evolution favors one genetic mechanism over another. While CDS mutations can enhance receptor sensitivity, they may also carry pleiotropic costs, such as potential lethality during development. In contrast, *cis*-regulatory modifications allow for a more flexible and context-dependent adjustment of gene expression, avoiding pleiotropic effects but leading to a more intermediate phenotype. The evolutionary balance between these mechanisms may reflect species-specific ecological demands, with specialists favoring CDS changes for heightened sensitivity and intermediates relying on *cis*-regulatory modifications for adaptive plasticity.

Consistent with this interpretation, *D. biarmipes*, which has diverged from *D. subpulchrella* and *D. suzukii* about 9 Ma [17], provides an informative outgroup for understanding the trajectory of $CO_2$-related adaptation. Our previous work showed that while *D. biarmipes* lays fewer eggs on rotten fruit compared to *D. melanogaster*, it does not exhibit the strong oviposition shift toward ripe fruit observed in *D. suzukii* and *D. subpulchrella* [16]. At the molecular level, *D. biarmipes Gr63a* shares the same amino acids as *D. melanogaster* at the three positions that differ between *D. suzukii* and *D. subpulchrella*, suggesting it may behave more similarly to *D. melanogaster* in the context of $CO_2$ attraction during egg-laying. Future studies examining *D. biarmipes* $CO_2$ responses will therefore be valuable for reconstructing the evolution of sensory adaptation along the lineage leading to the pest clade.

In our SSE experiments, we observed increased spike rates at $CO_2$ onset but no significant differences at baseline or steady state. This result aligns with known characteristics of olfactory receptors, where onset responses reflect transduction kinetics—such as odorant binding rates and channel activation dynamics—while steady-state responses result from the equilibrium between activation and inactivation processes [45,50]. Given that baseline and steady-state activity were similar across species, our findings suggest that $CO_2$ binding affinity remains unchanged, but that signal transduction occurs more rapidly in *D. suzukii* and *D. subpulchrella*. The increased transduction rate could arise from structural changes in the *D. suzukii* Gr63a protein, particularly at the C-terminal domain, which is extracellular in ligand-gated ion channels [51,52]. Mutations in this region may enhance binding efficiency or receptor activation dynamics, potentially contributing to the observed increase in $CO_2$ sensitivity [53,54]. Despite only three amino acid differences between *D. suzukii* and *D. subpulchrella* Gr63a proteins, the *D. subpulchrella* CDS alone was not sufficient to increase spike rate, reinforcing the hypothesis that both coding and *cis*-regulatory elements contribute to functional divergence.

Although both the *D. suzukii* CDS and *D. subpulchrella cis*-regulatory elements successfully increased $CO_2$ sensitivity in transgenic flies, these modifications did not alter oviposition behavior. Instead, transgenic flies continued to exhibit $CO_2$ aversion, consistent with wildtype *D. melanogaster* but distinct from *D. subpulchrella* and *D. suzukii* (Fig 5E). This suggests that an additional behavioral switch, acting downstream of the peripheral sensory system, modulates $CO_2$-driven oviposition preference.

Recent studies have demonstrated how higher-order circuits can reshape behaviors. For instance, species-specific courtship pheromone responses have been shown to be modulated by differential propagation of interneuron signaling in central neural circuits [34]. In addition, a recent study found that heat preference could be modulated by changes in the activation threshold of the peripheral gustatory receptor, Gr28b.d [55], but the switch from heat aversion to heat attraction

between species was ultimately driven by changes in the lateral horn. Another study found that attraction to noni fruit volatiles in *D. sechellia* was influenced by amino acid changes in odorant receptors that increase *D. sechellia*'s sensitivity to noni volatiles, yet the behavioral attraction itself was attributed to species-specific central projection patterns [21]. We believe that the $CO_2$ behavior valence during oviposition follows a similar structure. While we found that the sensitivity to $CO_2$ increased in *D. suzukii* and *D. subpulchrella* due to changes at the periphery, increased tolerance to higher $CO_2$ is most likely due to higher-order changes similar to those previously described. Therefore, the neural pathways governing $CO_2$ oviposition behaviors likely integrate sensory inputs with central processing mechanisms to generate species-specific behavioral responses.

Our results also provide insight into potential developmental constraints associated with heightened $CO_2$ sensitivity. The transgenic line carrying the *D. suzukii* CDS under the control of the *D. subpulchrella cis*-regulatory element (*DsubGr63acis-GAL4 > UAS-DsuzGr63acds*) was expected to have the highest $CO_2$ sensitivity, but at the time of experiments, we failed to generate homozygous individuals after many crosses. This suggests that excessive $CO_2$ sensitivity may disrupt larval development. Previous studies have shown that $CO_2$ exposure can immobilize larvae, halt cardiac function, and inhibit neuromuscular junction activity [56]. This raises the possibility that increased $CO_2$ sensitivity during development interferes with proper larval growth. In our case, we found that most vials containing the *DsubGr63acis-GAL4 > UAS-DsuzGr63acds* mutants had unhatched pupae. It is possible that *D. subpulchrella* pupae use $CO_2$ as a hatching cue and that $CO_2$ receptors are normally expressed at a later developmental stage in this species. Therefore, misexpression of $CO_2$ receptors in *D. melanogaster* larvae may induce physiological stress, preventing successful eclosion.

Together, our findings highlight the interplay between peripheral and central nervous system adaptations in shaping oviposition behavior. While *D. suzukii* and *D. subpulchrella* evolved peripheral modifications that increase $CO_2$ sensitivity, the persistence of $CO_2$ aversion in transgenic *D. melanogaster* suggests that behavioral changes require additional neural adaptations. Future research should investigate how sensory inputs are processed in central circuits to mediate species-specific oviposition behavior. These results provide a broader framework for understanding how different genetic mechanisms—CDS vs. *cis*-regulatory modifications—are selected in evolutionary contexts, balancing functional shifts with pleiotropic constraints. Understanding these mechanisms will provide deeper insights into the evolutionary dynamics of sensory adaptation and behavioral specialization.

## Methods

### General data information

All p-values are represented as *Q-values*, these are the corrected p-values. A Holm correction was used for all statistical tests. Custom R-scripts were used for all statistical analysis. Spikes were called and sorted with custom MATLAB scripts.

### Fly husbandry

All flies were reared on standard cornmeal medium at 24 °C, on a 12-h light-dark cycle (lights on at 8:00 AM). All behavioral experiments were conducted under the same conditions. For behavioral assays, we used: $w^{1118}$ for *D. melanogaster,* the wildtype strain for *D. suzukii* (SGD 17-5), and the wildtype *D. subpulchrella* strain (NGN6, Strain # E-15204, Ehime stock center). Gr63a[1] stock (BDSC #9941), the DmelGr63a[cis]-GAL4 (57659), the DmelGr63a[CDS]-UAS (23143), and the UAS-GFP strain (BDSC #67093) were obtained from Bloomington Drosophila Stock Center (Bloomington, USA).

### Fly crosses

The species-specific GAL4 lines and the Gr63a[1] lines are both on chromosome 3. To obtain a fly with both the GAL4 and Gr63a knockout, we created flies that were recombinant for both GAL4 and the Gr63a1 knockout using balancers.

We crossed the Gr63a^suzukii-GAL4, Gr63a^melanogaster-GAL4, and Gr63a^subpulchrella-GAL4 with the Gr63a[1] lines. Recombination events were confirmed by PCR and sequencing and were observed at a rate of 0.29. Primers for the Gr63a1 recombinant region used were Fwd: CGCCATTGACACAAATTCGAAA and rev: CGCTGACTTTGAGTGGAATGT. For the *D. subpulchrella* and *D. suzukii* GAL4 lines, the ATTB primers were used. Primers specific for the GAL4 region were used for *D. melanogaster* GAL4 (Fwd:GAACAACTGGGAGTGTCGCTA, Rev:TCCGATGAATGTCGCACT). These flies were then crossed to the balanced Gr63a UAS lines.

## Transgenic line generation

The UAS vectors were constructed by synthesizing the *D. subpulchrella* and *D. suzukii* Gr63a CDS and GFP protein, which were then cloned into the pJFRC7 UAS vectors using double digestion. These vectors were then injected into *D. melanogaster* embryos carrying the attP40 docking sites using the PhiC31 integrase system [57] All injections were performed by BestGene Inc. (Chino Hills, USA). For the GAL4 vectors, The *D. subpulchrella* and *D. suzukii* Gr63a regulatory elements were isolated by long-range PCR as previously described [40]. Briefly, forward primers (*D. suzukii*: GGGGACAAGTTTGTACAAAAAAGCAGGCTTATTGGCCAAATTGAAGGAGAG GT/ D. subpulchrella: GGGGACAAGTTTGTACAAAAAAGCAGGCTTATCCGGAGAGACTGTGTCCG) were designed for the region directly upstream of the predicted ATG and a reverse primer (*D. suzukii*: GGGGACCACTTTGTA-CAAGAAAGCTGGGTTTCCGGAGAGACTGTGTCCG/ *D. subpulchrella*: GGGGACCACTTTGTACAAGAAAGCT-GGGTTATATATTCAACTCCAGTTTTGGCCAAATTGAAG) ~2.6 Kb upstream. The regulatory elements were then cloned into the GAL4 vector pBPGUw using BP followed by LR cloning. These vectors were then injected into VK0027 docking site with *w^1118* background embryos (BDSC stock # 9744) using PhiC31 integrase and crossed with balancer lines.

## Quantitative RT-PCR

Virgin *D. suzukii*, *D. subpulchrella*, and *D. melanogaster* flies were collected and aged for 2–3 days. After 2–3 days, the flies were flash frozen in liquid nitrogen, and the antennae were dissected from around 130 female flies for each replicate. In total, 3 antennal samples were collected for each species. The RNA was extracted with custom protocols and DNAse treated, and oligo cleaned (Oligo Clean & concentrator kit cat#: 11-380B). The cleaned samples were reverse transcribed using the Takara kit (PrimeScript RT Reagent Kit (Perfect Real Time) cat # RR037B) and the qPCR was conducted using TBP as a control. The following primer sequences were used:

TBP *D. subpulchrella* and *D. suzukii* Fwd: CCACGGTGAATCTGTGCT

TBP *D. subpulchrella* and *D. suzukii* Rev: GGAGTCGTCCTCGCTCTT

TBP *D. melanogaster* Fwd: CCACGGTTAATCTGTGCTG

TBP *D. melanogaster* Rev: GGAGTCGTCCTCACTCTT

Gr63a *D. suzukii* Fwd: CCAACTACTATCGACGCAAGAAA

Gr63a *D. suzukii* Rev: CGCCGTACAAATATGCCTGG

Gr63a *D. subpulchrella* Fwd: TAGCACATGGCGTTTCCAGT

Gr63a *D. subpulchrella* Rev: GAAACACATTGGACCCGCTG

Gr63a *D. melanogaster* Fwd: AGGCATACTTGTACGGGGTC

Gr63a *D. melanogaster* Rev: ATGCGGATTACGAAGCTCCTC

PLOS Genetics

## Olfactometer chamber design

The four-field olfactometer size and shape were adapted from previous publications [44]. The height of the main arena was increased slightly (height = 1.2 cm) to accommodate an egg-laying substrate. The chamber is 30 cm wide (from one tip to another). The chamber has five layers, each layer of the chamber was laser cut on acrylic plastic. The bottom layer has an exit hole in the center to let $CO_2$ flow out of the chamber. The second layer outlines the area that is queried by the flies. It has a puckered star shape to amplify the formation of four distinct quadrants. Each corner in this layer has an input valve to which airflow can be attached. Above this layer is a neoprene layer, which creates a tight grip on the glass and creates an airtight chamber. All layers are pressed together with the top layer, in which screws hold the four layers together. The chamber was tested in a water bath to ensure that there were no leaks in the design.

## Oviposition behavior trials

For the 16-hours overnight oviposition trials, flies were collected as virgins and aged for 3–4 days in food vials. On day 3 or 4, the flies were mated and given yeast and an egg-laying substrate was withheld for a day. The next day, 12 females were placed into the four field olfactometer. The bottom of the chamber was filled with an agarose mixture that serves as an egg-laying substrate. Air from a controlled gas tank with atmospheric (400 ppm) $CO_2$ concentration was then dispensed into two inlets while air from another gas tank with 1% (10,000 ppm) $CO_2$ was dispensed into the other inlets (Fig 2A). The airflow was controlled at 15 ml/min$^{-1}$ using flowmeters. The 1% $CO_2$ concentration was chosen as it is 25 times higher than atmospheric levels of $CO_2$ but not high enough that the Ir25a pathway could be activated [58]. After 16 hours the experiment was ended, and the eggs on each half of the chamber were counted. An oviposition preference index was then calculated by subtracting the number of eggs found on the side with 0.04% $CO_2$ from the number of eggs found on the 1% $CO_2$ side and dividing this number by the total number of eggs.

For the overnight oviposition trials with control for air, 36 female flies from the same vials were aged for 3–4 days. On day 3 or 4, the flies were mated and given yeast to encourage egg-laying. Egg-laying substrates were withheld for 24 hours before the experiment. The same agarose mixture as previously described was placed on the bottom of the chamber. On the day of the trial, 3 chambers were prepared—one with 1% $CO_2$ vs 0.04% $CO_2$ (the original experiment), one with 0.04% $CO_2$ entering at all four inlets, and one without any air entering the chamber. After 16 hours, the experiment was ended, and the eggs in each chamber were counted.

## Positional preference trials

For the positional preference trials, mated adult flies were placed into the four-field olfactometer without an egg-laying substrate. Air with 1% $CO_2$ was dispensed into two adjacent quadrants of the chamber and air with 0.04% $CO_2$ into the two opposite quadrants. The number of flies in each $CO_2$ condition zone was recorded after 15 minutes. The $CO_2$ conditions were then permuted by switching the dispensing tubes and 15 mins later a new positional recording was made. A positional preference index was then calculated by subtracting the number of flies on the 0.04% $CO_2$ side from the 1% $CO_2$ and divided by the total number of flies. Each data point consisted in the average of the two 15-min positional preferences of each trial.

## Oviposition behavior trials with transgenic lines

Homozygous flies were collected as virgins and aged for 3–4 days in food vials. For the *DsubGr63a*-GAL4 lines and the line *DsuzGr63a*-GAL4 > DsubGr63a-UAS, we were unable to obtain a lot of homozygous flies, and we had to supplement these trials with heterozygous flies, making the comparison somewhat difficult. On day 3 or 4, the flies are mated and given yeast to encourage egg-laying. For each trial, 12–17 females are placed into the four field olfactometer. The trial was conducted as described in the Oviposition behavior trials section. The eggs per fly rate was calculated by dividing the number of eggs by the number of flies added to the chamber.

## Quantitative RT-PCR for transgenic lines used for SSE recordings

For each of the nine transgenic lines used for SSE recordings, we collected 15 adult females from the transgenic lines on the morning of eclosion and phenotyped them to obtain homozygote strains for both the GAL4 and UAS loci. For seven of the nine transgenic lines, we obtained homozygote individuals for dissections, but for the remaining two, we were unable to collect sufficient samples thus collected heterozygote UAS (or GAL4) and homozygote GAL4 (or UAS). Dissections were performed on the flies 2–3 days after collection. The females were first anesthetized with $CO_2$ and had their heads removed with a sterile razor blade. Five heads constituted a single sample and upon dissection were placed in an Eppendorf tube and submerged in liquid nitrogen and stored in a -80 ° C until all samples were collected and ready for quantitative RT-PCR. Total RNA was extracted using a standard TRIzol protocol followed by DNase treatment, and RNA concentration and purity were assessed with a NanoDrop spectrophotometer. Reverse transcription was performed using the PrimeScript RT Reagent Kit (TaKaRa, Cat#RR037A). Quantitative PCR was performed with PowerUp SYBR Green Master Mix (ThermoFisher, Cat#A25741) to measure expression of the target gene Gr63a, using an identical primer pair to amplify a region with the three coding variants. The ribosomal protein gene Rpl32 served as an internal reference for normalization. Fluorescence signals were detected on a QuantStudio 5 Real-Time PCR system, and relative expression levels were calculated using the ΔCt method, with the homozygous CDS-MEL, cis-MEL line as the baseline.

## Single sensillum electrophysiology

*D. subpulchrella* (NGN6) *D. suzukii* (SGD 17-5), and *D. melanogaster* ($w^{1118}$) females were anesthetized on ice for two minutes and placed into a cut pipette tip. The head and body were stabilized in the pipette tip using hot wax. UV glue was used to glue the arista and 2nd antennal segment. The fly was then transferred to an upright compound microscope with a 50 × air objective. Glass electrodes were pulled and filled with saline and a silver chloride electrode. The reference electrode was placed into the eye of the fly, while the recording electrode was placed into the lymph of the ab1C sensillum. The ab1C neuron is in a large basiconic sensilla, which is easy to distinguish from other sensilla as it has 4 spike sizes at rest. A $CO_2$ stimulus (2% $CO_2$) was puffed for 2 second intervals close to the fly head with 10 second rests between replicate recordings. Electrical signals were acquired using a Model 2400 amplifier (A-M Systems). Spikes were called and counted using a custom MATLAB script. For each species, 10 different ab1C sensilla were recorded and spikes were called before, during and after $CO_2$ onset. For the transgenic line recordings, the study was conducted in a deidentified manner. The identities of the lines were disclosed only after spike sorting and calling were completed. The onset spike rate was compared between groups using a one-way ANOVA followed by a TukeyHSD post hoc test. The resulting 36 pairwise comparisons were summarized using the multcompView [59] package, which assigns letters to each group to indicate statistical similarity. Groups that share at least one letter (for example, "ab" and "a") are not significantly different, whereas groups that do not share any letters (for example, "a" versus "b") differ significantly.

## Interpreting the heterozygous transgenic line in SSE experiments

To compare the heterozygous line, DsubGr63a$^{cis}$-GAL4 > UAS-DsuzGr63a$^{cds}$, with the homozygous lines during analysis, we employed a modeling approach (see methods) to predict the expected homozygous spike rate for this individual. The proportions of our known homozygous lines were used to extrapolate the predicted homozygous spike rate of DsubGr63a$^{cis}$-GAL4 > UAS-DsuzGr63a$^{cds}$. We calculated the average spike rate proportion between the *D. suzukii* and *D. subpulchrella* CDS driven by the *D. melanogaster* or *D. suzukii* GAL4. Then we averaged the two scaling factors together and multiplied 5 randomly selected data points from the *D. subpulchrella* CDS driven by the *D. subpulchrella* enhancer to get our first estimate. We then repeated these steps with the *D. suzukii* and *D. melanogaster* CDS driven by the *D. melanogaster* or *D. suzukii* cis-regulatory element. We then used this second scaling factor to multiply 5 randomly selected data points from the *D. subpulchrella* cis-regulatory element driving the *D. melanogaster* CDS. As a control to access the

robustness of our prediction, we used a similar approach to estimate the spike rate of the *D. subpulchrella cis*-regulatory element driving the *D. melanogaster* CDS and compared it to the measured value.

### Immunohistochemistry

The *Gr63asubpulchrella*-GAL4*, Gr63asuzukii*-GAL4*, and Gr63amelanogaster*-GAL4 flies were crossed with UAS-GFP (BDSC#67093) lines. The brains of homozygous female flies were dissected in 1X-PBS. Brains were pre-fixed in 4% PFA in 1X PBS and fixed in 4% PFA with 0.25% TritonX-100. Brains were degassed using a vacuum chamber to remove the gas stored in the trachea and blocked in blocking buffer (1% NGS, 0.02% NaN3, Triton X-100, 1X-PBS) for at least 2 hours. Brains were labeled with primary mouse-nc82 antibody (1:30) (DSHB Hybridoma Product nc82) and chicken-gfp antibody (1:1000) (Rockland, 600–901-215) for 2 days. The brains were then incubated with secondary antibodies—goat anti-chicken (1:800) (Alexa Fluor 488) (Thermo Fisher, Catalog # A-11039) and Goat anti-mouse (1:200) (Alexa Fluor 633) (Thermo Fisher, Catalog # A-21052)—overnight. Brains were mounted on glass slides in FocusCLear (catalog number: FC-101) and imaged on a confocal microscope at 10X magnification.

### Supporting information

**S1 Fig. Single sensillum electrophysiology recordings for the ab1c neuron in wildtype flies.** A-B) The maximum spike rate across each ab1c neuron recording is significantly higher in *D. suzukii* and *D. subpulchrella* compared to *D. melanogaster* (Kruskal-Wallis test followed by pairwise Wilcoxon signed-rank test with Holm correction, $Q < 0.001$, $Q = 0.017$). C-D) The ab1c spike rate of the three species is not significantly different at baseline (Kruskal-Wallis test followed by pairwise Wilcoxon signed-rank test with Holm correction, Dsub-Dmel: $Q = 1.00$, Dsuz-Dmel: $Q = 1.00$, Dsub-Dsuz: $Q = 0.89$). E-F) The ab1c spike rate of the three species is not significantly different at steady state, which was calculated across 1000 ms (Kruskal-Wallis test followed by pairwise Wilcoxon signed-rank test with Holm correction, Dsub-Dmel: $Q = 0.066$, Dsuz-Dmel: $Q = 1.00$, Dsub-Dsuz: $Q = 0.056$).
(PDF)

**S2 Fig. Species-specific effects of Gr63a coding and regulatory variation on $CO_2$-evoked neural activity.** A) LFP recordings for each transgenic line of the ab1c neurons recorded in Fig 4. Each grey line is the average of five replicates from one ab1c neuron, and the colored lines are the average across ten neurons. B) Without correcting for the heterozygous individual, the slope of the LFP at $CO_2$ onset was calculated across 750 ms. For the *CDS,* transgenic lines with *D. suzukii* CDS had the highest rate of change, followed by *D. melanogaster* and then *D. subpulchrella* (Kruskal-Wallis test followed by pairwise Wilcoxon signed-rank test with Holm correction, Dsub-Dsuz: $Q = 0.00027$, Dsuz-Dmel: $Q = 0.01990$, Dsub-Dmel: $Q = 0.01836$). For the *cis*-regulatory element, the *D. subpuclrehlla* had the highest rate of change and trended lower than the other species elements (Kruskal-Wallis test followed by pairwise Wilcoxon signed-rank test with Holm correction, Dsub-Dsuz: $Q = 0.092$, Dsuz-Dmel: $Q = 0.982$, Dsub-Dmel: $Q = 0.092$).
(PDF)

**S3 Fig. Comparison of *Gr63a* coding sequences and spike rates of transgenic lines.** A) Comparison of *Gr63a* coding sequences among *D. suzukii*, *D. subpulchrella*, and *D. melanogaster.* B) Predicted spike rates at CO2 onset for the SUB-GAL4 > UAS-SUZ individual as well as the SUB-GAL4 > UAS-SUB control. The predicted spike rate of SUB-GAL4 > SUB-UAS and the actual spike rate are very similar (Kruskal-Wallis test followed by pairwise Wilcoxon signed-rank test with Holm correction, Q = 0.19, mean of B = 224, mean of estimated B = 228), which gave us confidence that the predicted homozygous spike rate for SUB-GAL4 > UAS-SUZ is robust. C) Heat map comparing the spike rate between homozygous transgenic lines, including the modeled SUB-GAL4 > UAS-SUZ value, carrying *subpulchrella* and *melanogaster* proteins with *cis*-regulatory sequences from the three species. The *D. suzukii* CDS (Kruskal-Wallis test followed by pairwise Wilcoxon signed-rank test with Holm correction, Dmel–Dsuz: $Q = 1.1e-08$ and Dsub-Dsuz: $Q = 6.3e-07$) and *D.*

*subpulchrella cis*-regulatory element (Kruskal-Wallis test followed by pairwise Wilcoxon signed-rank test with Holm correction, Q = 0.0139 between *D. melanogaster* and *D. subpulchrella* and Q = 0.004 between *D. subpulchrella* and *D. suzukii*) resulted in the highest spike rate. D) Spike rate comparisons of homozygote transgenic lines carrying *subpulchrella* and *suzukii* proteins with regulatory sequences from the three species. E) Spike rate comparisons of homozygote transgenic lines carrying *melanogaster* and *suzukii* regulatory sequences with CDS from the three species.
(PDF)

**S4 Fig. Quantitative RT-PCR of the transgenic lines and their correlation with average spike rates.** A) qRT-PCR of the transgenic lines, normalized to the line containing homozygote MEL-GAL4 and UAS-MEL. Note that we were not able to obtain homozygous flies for two strains at the time of the experiment. B) Correlation between average spike rate and expression levels for the six lines that have homozygous flies for both quantitative RT-PCR and SSE recordings.
(PDF)

**S1 Table. Raw Ct values for the antennal qPCR for each species.** For each species, we used three technical replicates for three biological replicates for both *Gr63a* and the endogenous control TBP.
(DOCX)

## Acknowledgments

We thank all members of the Zhao and Nagel lab for helpful discussions and technical feedback, and critical feedback from Vanessa Ruta, Leslie Vosshall, Rory Coleman, and Philipp Brand. We thank the Rockefeller University's Precision Instrumentation Technologies Resource Center, especially Jim Petrillo and Peer Strogies for help designing and building the four-field olfactometer, and The Rockefeller University Bio-Imaging Resource Center for training and advice on confocal imaging (RRID:SCR_017791). We also thank Fabian Gadau for help designing the four-field olfactometer, Sachin Sethi and Audrey L. Francis for sharing their IHC protocol and their helpful discussions.

## Author contributions

**Conceptualization:** Alice Gadau, Katherine I. Nagel, Li Zhao.

**Data curation:** Alice Gadau, Sasha Mills, Xin Yu Zhu Jiang, Nicolas Svetec, Ziyu Xu, Katherine I. Nagel.

**Formal analysis:** Alice Gadau, Sasha Mills, Xin Yu Zhu Jiang.

**Funding acquisition:** Alice Gadau, Li Zhao.

**Investigation:** Alice Gadau, Sasha Mills, Cong Li, Nicolas Svetec, Ziyu Xu, Katherine I. Nagel, Li Zhao.

**Methodology:** Alice Gadau, Cong Li, Nicolas Svetec, Wanhe Li, Katherine I. Nagel.

**Project administration:** Katherine I. Nagel, Li Zhao.

**Resources:** Nicolas Svetec, Katherine I. Nagel, Li Zhao.

**Supervision:** Nicolas Svetec, Wanhe Li, Katherine I Nagel, Li Zhao.

**Visualization:** Alice Gadau, Cong Li.

**Writing – original draft:** Alice Gadau, Li Zhao.

**Writing – review & editing:** Alice Gadau, Li Zhao.

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
