## [Decision Letter · Decision Letter 0]

2 Jul 2025

PGENETICS-D-25-00456

Divergent molecular processes drive convergent behavioral innovation in the Drosophila suzukii species complex

PLOS Genetics

Dear Dr. Zhao,

Thank you for submitting your manuscript to PLOS Genetics. After careful consideration, we feel that it has merit but does not fully meet PLOS Genetics's publication criteria as it currently stands. Therefore, we invite you to submit a revised version of the manuscript that addresses the points raised during the review process.

Please submit your revised manuscript within 60 days Aug 31 2025 11:59PM. If you will need more time than this to complete your revisions, please reply to this message or contact the journal office at plosgenetics@plos.org. Please include the following items when submitting your revised manuscript:

We look forward to receiving your revised manuscript.

Kind regards,

Emily L Behrman

Guest Editor

PLOS Genetics

Justin Fay

Section Editor

PLOS Genetics

Aimée Dudley

Editor-in-Chief

PLOS Genetics

Anne Goriely

Editor-in-Chief

PLOS Genetics

**Additional Editor Comments:**

Thank you for submitting your manuscript entitled “Divergent molecular processes drive convergent behavioral innovation in the Drosophila suzukii species complex” for consideration at PLOS Genetics. We appreciate the work that you have accomplished and agree that this manuscript has the broad scientific implications appropriate for the audience of PLOS Genetics. Four experts in the field have evaluated the manuscript and provided feedback. All four reviewers were enthusiastic about the general results but pointed out several points that require additional attention.

We request text revisions that temper the causal link between the molecular and neural processes and the behavioral variation studied here to a level commensurate with the results presented in the manuscript.

Reviewers brought up several concerns regarding the regulatory differences in the Gr36a that should be addressed, including incorporation of what is known about regulation in other species and a broader comparison of regulatory and coding sequence divergence among additional species of intermediate divergence between the species studied here in the D. suzukki complex and D. melanogaster. The concern regarding the how the genomic location differences in transgenes and different UAS vectors may confound comparison of expression levels within D. melanogaster is of particular importance for the conclusions of this manuscript and would be addressed by the request for quantification of Gr63a transcript abundance.

Several reviewers questioned the necessity of the model of single sensillum recordings and suggest that the narrative may be clarified if the model is moved to the supplemental material.

Justification for experimental conditions and demonstration if the species exhibit the same behaviors at lower, more ecologically relevant levels of CO2.

The reviewers additionally provided useful comments that would help clarify portions of the manuscript.

**Journal Requirements:**

At this stage, the following Authors/Authors require contributions: Alice Gadau, Xin Yu Zhu Jiang, Sasha Mills, Nicolas Svetec, Ziyu Xu, Wanhe Li, Katherine I Nagel, and Li Zhao. Please ensure that the full contributions of each author are acknowledged in the "Add/Edit/Remove Authors" section of our submission form.

The list of CRediT author contributions may be found here: https://journals.plos.org/plosgenetics/s/authorship#loc-author-contributions

https://journals.plos.org/plosgenetics/s/submission-guidelines#loc-parts-of-a-submission

4) We do not publish any copyright or trademark symbols that usually accompany proprietary names, eg ©,  ®, or TM  (e.g. next to drug or reagent names). Therefore please remove all instances of trademark/copyright symbols throughout the text, including:

- TM on page: 20.

5) Please upload all main figures as separate Figure files in .tif or .eps format. For more information about how to convert and format your figure files please see our guidelines:

6) We notice that your supplementary Figures, and Tables are included in the manuscript file. Please remove them and upload them with the file type 'Supporting Information'. Please ensure that each Supporting Information file has a legend listed in the manuscript after the references list.

7) Some material included in your submission may be copyrighted. According to PLOSu2019s copyright policy, authors who use figures or other material (e.g., graphics, clipart, maps) from another author or copyright holder must demonstrate or obtain permission to publish this material under the Creative Commons Attribution 4.0 International (CC BY 4.0) License used by PLOS journals. Please closely review the details of PLOSu2019s copyright requirements here: PLOS Licenses and Copyright. If you need to request permissions from a copyright holder, you may use PLOS's Copyright Content Permission form.

Potential Copyright Issues:

- Figures 1 and 5. Please confirm whether you drew the images / clip-art within the figure panels by hand. If you did not draw the images, please provide (a) a link to the source of the images or icons and their license / terms of use; or (b) written permission from the copyright holder to publish the images or icons under our CC BY 4.0 license. Alternatively, you may replace the images with open source alternatives. See these open source resources you may use to replace images / clip-art:

8) We note that your Data Availability Statement is currently as follows: "All included in the manuscript.". Please confirm at this time whether or not your submission contains all raw data required to replicate the results of your study. Authors must share the “minimal data set” for their submission. PLOS defines the minimal data set to consist of the data required to replicate all study findings reported in the article, as well as related metadata and methods (https://journals.plos.org/plosone/s/data-availability#loc-minimal-data-set-definition).

- The points extracted from images for analysis..

9) Please amend your detailed Financial Disclosure statement. This is published with the article. It must therefore be completed in full sentences and contain the exact wording you wish to be published. State what role the funders took in the study. If the funders had no role in your study, please state: "The funders had no role in study design, data collection and analysis, decision to publish, or preparation of the manuscript.".

**Reviewers' comments:**

Reviewer's Responses to Questions

**Comments to the Authors:**

Reviewer #1: Gadau et al. investigate the genetic basis of adaptation by examining how CO2 influences oviposition behavior and sensory neuron activity across three Drosophila species. Using a combination of behavioral, electrophysiological, and genetic approaches, they argue that convergent oviposition traits have evolved through distinct genetic mechanisms—a coding sequence change in one species and a cis-regulatory change in another. The authors developed a refined oviposition assay and demonstrated that CO2 differentially affects both oviposition site preference and egg-laying quantity among species. Single sensillum recordings from CO2-sensitive ab1c neurons expressing GR63a revealed species specific differences in CO2 sensitivity. To dissect the roles of coding versus regulatory changes, the authors generated new transgenic lines that allow independent manipulation of each. These tools showed that a GR63a coding change in one species and a GR63a regulatory change in another species enhance ab1c neuron sensitivity to CO2. Finally, they examined the respective contributions of coding and regulatory changes to oviposition behavior, finding that distinct genetic changes can converge on similar egg-laying suppression traits.

The authors demonstrate a striking cross-species variation in ab1c neurons’ CO2 sensitivity, and the data support the idea that this variation results from convergent evolution of coding and cis-regulatory sequences. However, the relevance of these neuronal changes to behavioral variation remains unclear. The authors show that oviposition behavior differs between D. melanogaster and both D. subpulchrela and D. suzukii in two key ways: only D. melanogaster avoids 1.0% CO2 as an oviposition site, and only D. melanogaster suppresses egg-laying in response to 1.0% CO2. As the authors acknowledge in the Results and Discussion, changes in ab1c neuron CO2 sensitivity do not explain differences in either oviposition site preference or egg-laying suppression. Consequently, the genetic basis for the observed behavioral divergence across species remains unresolved. Thus, the current data fall short of supporting the manuscript’s central claim—highlighted in the Title and Abstract—that divergent molecular processes drive convergent behavioral traits. In my view, the most compelling finding of the manuscript is the interspecies differences in ab1c neuron CO2 sensitivity of ab1c neurons and its underlying genetic mechanisms, even though the behavioral data suggest that these neural differences are not directly linked to the behavioral variation studied here.

In the transgenic experiments, the authors carefully considered heterozygosity versus homozygosity to ensure fair comparisons. However, two potentially important differences exist between the D. melanogaster transgenic lines and those used for D. subpulchrela and D. suzukii. First, the Gal4/UAS constructs were inserted at different genomic loci: DmelGr63acis-GAL4 and DmelGr63aCDS-UAS were randomly inserted (Weiss et al., 2011, Neuron; Jones et al., 2007, Nature), whereas the D. subpulchrela and D. suzukii constructs were targeted to defined sites (attP2, attP40, or VK00027). Because genomic position can strongly influence transgene expression levels, this difference complicates cross-species comparisons. Second, the UAS vectors differed: DmelGr63aCDS-UAS used the pUAST vector with 5xUAS (Jones et al., 2007, Nature), while the other constructs used pJFRC7 with 20xUAS, likely resulting in different expression levels. These differences make it complicated to determine whether the observed variation in CO2 sensitivity truly reflects differences in coding sequences or cis-regulatory elements.

In the analysis of single sensillum electrophysiology data, the authors estimated the responses of homozygous lines through extrapolation. However, this approach was validated under only one condition (cis-SUB > CDS-SUB), and it remains unclear whether the predicted spike rate for cis-SUB > CDS-SUZ—which is substantially higher than any observed response—is plausible. Since the same conclusion is supported even without these extrapolated values (Fig. S3B, C), the necessity of including this modeling is unclear.

In the analysis of behavioral data from transgenic lines, it is unclear how the authors addressed heterozygosity versus homozygosity. Are the conclusions consistent when comparisons are limited to homozygous flies? Additionally, it is not explained why obtaining homozygotes was more challenging for the behavioral experiments than for the single sensillum electrophysiology experiments.

Line 196: “found that the substantial number of substitutions in the amino acid sequence have evolved at the c-terminus.”

Please indicate the C-terminal region in Fig. S3A so that readers can easily confirm this claim. It would also be helpful to discuss the potential functional implications of the selective accumulation of mutations in the C-terminal region.

Fig. 3B

The blue and orange highlights are difficult to see. Most of the sequence appears in pink and red. Also, it is challenging to evaluate from this panel whether the majority of the changes are clustered in the C-terminus.

Line 198: “D. subpulchrella and D. suzukii only differ 199 from each other at two amino acid sites: sites 48 and 450”

These two changes appear to underlie the differential CO2 sensitivity between CDS-SUB and CDS-SUZ. Are both changes located in the C-terminus? This specificity is striking and may deserve further discussion in the text, particularly regarding how such minimal changes could yield substantial functional differences.

Line 249: “However, this finding was not corroborated by the spike rate data, and it may be attributed to the contribution of nearby neurons when calculating LFP.”

Does this imply that the Gal4 driver used for expressing Gr63a may have also driven the expression in neurons adjacent to ab1c, with different CO2 sensitivities between D. melanogaster and D. subpulchrella CDS? As currently written, it is difficult to understand what mechanism the authors have in mind for the discrepancy between LFP and spike data.

Reviewer #2: In their manuscript entitled “Divergent molecular processes drive convergent behavioural innovation in the Drosophila suzukii species complex” Gadau and colleagues describe a differential sensitivity to CO2 between three Drosophilid species (suzukii, melanogaster, and the intermediate-phenotype subpulchrella) as arising through two distinct pathways: alteration of the coding sequence of a receptor (Gr63a) and the evolution of a cis-regulatory element controlling expression of the same receptor. This builds on previous work that identifies the CO2 receptor (Gr63a) as being differentially expressed in subpulchrella and the subject of positive selection in suzukii.

In general the experiments are well controlled and described, the data are clearly presented, and the interpretation of said data is conservative. The implications for the evolution of novel behaviours are clearly interesting and important. I have a few comments below for the authors that might strengthen the arguments they put forward, particularly surrounding the interpretation of qPCR results from wild-type flies of three species then extrapolated to transgenic expression experiments using the Gal4 > UAS system with three different CREs.

Major comments:

- the finding that high levels of CO2 seemed to suppress egg-laying in melanogaster is intriguing, though I struggle with the choice of 1% CO2 in this assay as the authors themselves state that ripe fruit is more typically associated with ~0.16% CO2 (nearly an order of magnitude less). are the flies just generally less active at such high concentrations? is this known? is 1% standard? It seems important to test egg-laying behaviour at lower levels of CO2 to see if the phenotypes observed here are in the realm of possibility for flies experiencing ripe fruit as opposed to piped in gas in an olfactometer at much higher concentrations.

- The increase in gene expression of Gr63a (as measured by transcript abundance) in subpulchrella could be a result of (at least) two factors: an increase in transcript abundance within the same cells, or expression of 63a in novel cells. The expression pattern showing innervation of the V-glomerulus in all conditions seen in Figure 5 appears to make the second option unlikely, and the lethality observed with the sub-CRE and the suz-CDS appears to suggest that the CRE does in fact drive differential levels of transcript abundance in a transgenic melanogaster context. However, the transcript abundance in the transgenic context is not quantified (e.g. by qPCR) and given that there is the potential for additional amplification from the Gal4 > UAS system, it would be important to see these numbers for the 9 transgenic combinations tested in figures 4 and 5 so that we can compare them to the wild-type abundances observed for the three species.

Minor comments:

- transcript abundance (as measured by qPCR) does not necessarily predict protein abundance, and I would like to see that issue explicitly acknowledged and discussed in the text… in addition, do we know for sure that increased levels of receptor protein complexes leads to increased sensitivity? It seems like an ok assumption, but I’m not aware of any studies making this link explicitly. This all needs to be discussed.

- line 302 misplaced reference 16

- Line 201 - UAS is not a gene - this section should be tightened up as creating plasmids is not the same as creating transgenic flies and it is difficult to determine exactly what was done here. The methods section starting line 413 did not clarify exactly where these lines ended up - Attp2? Attp40? Why are multiple landing sites mentioned, and which constructs ended up where?

- Line 215 should read ‘then crossed’ - in general the figure legend for this figure is not clearly described

Reviewer #3: A controversial question in 21st century evolutionary developmental biology has been whether regulatory changes or coding sequence changes drive interspecific phenotypic divergence. In this paper, Gadau and colleagues examine a gustatory receptor in two closely related pests (D. suzukii and D. subpulchrella) and claim that both processes have played a role in the evolution of a novel response to carbon dioxide, with coding sequence changes predominating in D. suzukii and regulatory changes in D. subpulchrella.

While the results are intriguing, I have questions about how the study was conducted and the authors' interpretation of their findings:

1) The authors make no mention of non-pest species that are more closely related to D. suzukii and D. subpulchrella than the model organism D. melanogaster. What is the response of a fly such as D. biarmipes, for example, to carbon dioxide? Where does this species prefer to lay its eggs in their oviposition experiments? If the authors have not carried out these experiments, they could at least use the D. biarmipes genome to examine how the gustatory receptor they focus on (Gr63a) diverged prior to the evolution of the pest clade. Furthemore, they should mention the importance of examining D. biarmipes and other closely related species in the Discussion.

2) I find it interesting (and possibly surprising) that the results for the D. suzukii and D. subpulchrella Gr63a coding sequence differ dramatically when there are only two amino acid sequence differences (line 199). I believe that examining the D. biarmipes sequence could shed light on this issue.

3) From the Methods section, it appears that the authors assumed that the regulatory region for Gr63a, in each species, was in the 1 kb upstream of the start codon (lines 417 to 426). This is a dubious assumption, however, and I believe they should discuss what is known in D. melanogaster (and possibly other species) about the regulation of this gene. Furthermore, do the regulatory regions they identified contain binding sites for upstream transcription factors? How diverged are these regulatory regions between D. suzukii and D. subpulchrella?

4) I find these results surprising, given how closely related these two species are. I think the authors should carefully consider if there are any alternative explanations for their findings and mention these in the Discussion.

5) In a few cases (e.g. lines 255 to 262) the authors have including text in the Results that would be more appropriate in the Discussion.

6) I was confused by the legend to Figure 5. There does not appear to be an explanation for panels A and B.

Reviewer #4: In this work, Gadau et al. address the genetic and neural basis of behavioral adaptation in D. suzukii, a major agricultural pest, and its sister species, D. subpulchrella. In parallel with their behavioral shift toward laying eggs in ripe fruits that emit elevated levels of CO₂, the authors establish that both species exhibit a lack of CO₂ avoidance during egg laying compared to D. melanogaster. Using single sensillum electrophysiology, the authors demonstrate that in both species, the relevant sensory neurons ab1C, which express the CO₂ receptor Gr63a, are in fact more sensitive to CO₂. Motivated by evidence of positive selection on Gr63a in D. suzukii and higher Gr63a expression in D. subpulchrella, the authors tested the effects of both protein-coding and cis-regulatory sequences of Gr63a from D. suzukii, D. subpulchrella, and D. melanogaster using chimeric transgenes expressed in a D. melanogaster Gr63a null background. They show that the coding sequences of D. suzukii and the cis-regulatory sequences of D. subpulchrella both enhance CO₂ sensitivity in ab1C neurons. Further behavioral assay testing CO₂ preference during egg laying did not reveal a corresponding behavioral effect. Overall, this work is highly significant: it pinpoints the precise genetic mechanisms in neural evolution for an agriculturally and economically relevant behavioral shift. Notably, while the role of protein-coding changes in sensory receptor evolution is well appreciated, this study provides further novel insights by revealing divergent genetic mechanisms in the same gene drives similar, behaviorally relevant neural changes. The quality of the work is overall excellent, and the transgenic experiments are particularly systematic and rigorous. While the data are strong and broadly support the major conclusions, I have a few suggestions that primarily pertain to the precise interpretation of some results.

1. The authors have addressed species differences at two levels: the electrophysiological level and the behavioral level (both general avoidance and avoidance in the context of oviposition). The genetic mechanisms underlying evolutionary changes in electrophysiological properties did not produce behavioral effects that align with wild-type phenotypes. This result is well expected. If the ab1C neurons in D. suzukii and D. subpulchrella exhibited decreased CO₂ sensitivity, a corresponding reduction in behavioral avoidance could be attributed to changes at the peripheral sensory level. However, in this case, since D. suzukii and D. subpulchrella show increased CO₂ sensitivity, their lack of CO₂ avoidance during egg laying most likely arises from changes at a higher-order processing level that assigns valence to CO₂ signals. Based on this, I have a few suggestions: First, the authors indeed provided a thoughtful discussion and interpretation of their results in the first paragraph of the Discussion section. They may also refine the wording in earlier sections to avoid the impression that increased CO₂ sensitivity is expected to result in reduced CO₂ avoidance, which might confuse readers. Second, while the increased CO₂ sensitivity in D. suzukii and D. subpulchrella may very well be related to their behavioral adaptation of egg laying on ripe fruit, this connection has not yet been experimentally demonstrated, and the relevant ecological and behavioral context remains unclear (e.g., it is possible that the relevant behavioral phenotypes are in larvae rather than in adults). The authors may tone down their language to avoid potentially premature inference (e.g., Title, Abstract lines 23–24). Finally, the discussion of the contribution of processes “downstream of the peripheral sensory system” (lines 346–348) could be further developed. For example, consider citing Capek et al. 2025 (Nature) and Dürr et al. 2025 (Cell Reports), in addition to Seeholzer et al. 2018. Capek et al. 2025 is particularly relevant to this conceptual discussion.

2. Protein-coding changes versus cis-regulatory changes: The authors have framed this study within the conceptual framework of evolutionary change through cis-regulatory versus protein-coding sequence alterations in the context of pleiotropic effects. While this framework has been highly influential in studies of phenotypic evolution, I do not think it is fully applicable here. Sensory receptor genes—such as Gr63a, the focus of this study—are typically not pleiotropic in the same sense as developmental master regulators, from which this framework originally derives. Instead, they tend to be uniquely specialized, with restricted expression patterns and dedicated functions. It is in theory possible that cis-regulatory changes might avoid pleiotropic effects if Gr63a is upregulated in specific developmental stage, but this paper has not demonstrated any evidence to suggest that cis-regulatory changes here are less pleiotropic. Additionally, I found the inference that protein-coding changes may have stronger phenotypic effects than cis-regulatory changes (e.g., lines 340–341) not particularly informative, as the relative impact of these changes highly depends on the nature of specific mutations. I suggest removing this interpretation or tune it down. I shall emphasize that regardless, it remains valuable to examine how similar or distinct genetic mechanisms underlie phenotypic evolution. In my view, this is where the primary conceptual contribution of this paper lies: demonstrating that distinct genetic changes within the same gene can drive ecologically relevant convergence of neural electrophysiological traits in two closely related species.

3. The transgene experiments are slightly confounded by the inability to generate viable homozygous adults of cis-SUB>CDS-SUZ due to lethality. The authors employed a modeling approach to predict the expected homozygous phenotype (spike rate). I don’t think the modeling is necessary for this paper: first, my intuition is that if there are any non-additive interactions between cis-SUB and CDS-SUZ, there is no available “truth” to model from. In another word, this prediction cannot extract information beyond what is already available and therefore does not provide any new or necessary knowledge; second, this predicted result is not critical for the main conclusion (the effect of cis-SUB and CDS-SUZ). Admittedly, I am not an expert in modeling and may not be best positioned to judge its suitability; other reviewers may be in a better position to weigh in. I also found that the result of SUB-GAL4>UAS-SUZ/CyO does not add much, given the differences in gene dosage and the complication of genetic background. I suggest removing these data or moving them to the supplementary data. Relatedly, the lethality of this homozygote is interesting. In the relevant discussion (lines 374–378), it might be worth mentioning that in D. melanogaster, Gr63a is also expressed in larvae and may contribute to CO₂ detection (co-expressed with Gr21a)—see Kwon et al. 2007 PNAS.

4. “A stepwise evolutionary system”: For me, “stepwise evolution” implies evolutionary changes occurring in successive steps along a mutational path (e.g., Karageorgi et al. 2020 Nature would be a good example). “Stepwise” might therefore be a confusing term here, given that the mutations (protein-coding and cis-regulatory) suggest independent/parallel (not successive) adaptation.

5. Expression level of Gr63a: does the higher expression level of Gr63a revealed by qPCR reflect an increase in gene expression per ORN, or an increase in the number of Gr63a-expressing ORNs (Fig. 3A)? I understand that this may be difficult to quantify directly due to the lack of genetic reagents in D. suzukii and D. subpulchrella, but perhaps the authors could quantify the fluorescence signal of the transgene mel/suz/sub-cis>GFP? At minimum, the authors should acknowledge both possibilities if they cannot distinguish between them, especially considering that expanding the number of sensory neurons is a common mechanism (e.g., Takagi et al. 2024 Nat Commun as a recent example in Drosophila).

6. When reporting species differences in CO₂ sensitivity of ab1C neurons, the authors should cite Krause Pham and Ray 2015 Scientific Reports, which first demonstrated higher CO₂ sensitivity in D. suzukii.

7. In places, I find the use of transition words (such as “but,” “however,” “surprisingly”) excessive, disrupting the logical flow (e.g., lines 92–96; lines 286–290). I suggest that the authors revisit the usage of these words and evaluate whether an emphasis on contrast or exception—which can be subjective—is truly necessary.

8. Line 301: D. subpulchrella, D. suzukii, and D. melanogaster Gr63a converge on the V-glomerulus. Given the specific meaning of “converge” in evolutionary biology, I suggest avoiding this word (also, genes don’t converge on glomeruli). I also think that this piece of data should appear earlier as a validation that the chosen cis-regulatory sequences work as expected in the transgene experiment.

9. Figure 4B: add statistical results for all pairwise comparisons (can use letters to denote significance) because panel C does not fully reveal this information. Similarly 5A and 5B.

10. A rigorous proof-reading is needed. Here are a few examples:

• Line 57: “significantly highly express” → “significantly highly expressed”

• Line 80: “CO2 - CO₂.”

• Line 84: “showing the first signs of a behavioral shift” → “suggesting a behavioral shift in CO₂ response”?

• Line 82: “like the behavior explained previously” → “consistent with previous reports”?

• Line 96: “avoidanec” → “avoidance.”

• Line 302: citation “16” was placed at the beginning of the sentence.

• Figure 1: panels C and D—make Y axis labels consistent.

• Most places: “ab1c neurons”; some places: “ab1C neurons”—make usage consistent.

**Have all data underlying the figures and results presented in the manuscript been provided?**

Reviewer #1: None

Reviewer #2: Yes

Reviewer #3: None

Reviewer #4: None

PLOS authors have the option to publish the peer review history of their article (what does this mean?). If published, this will include your full peer review and any attached files.

Reviewer #1: No

Reviewer #2: No

Reviewer #3: No

Reviewer #4: No

**Figure resubmission:**
---

## [Editor Report · Decision Letter 1]

7 Jan 2026

Dear Dr Zhao,

We are pleased to inform you that your manuscript entitled "Molecular evolution of CO2-sensing ab1C neurons underlies divergent sensory responses in the Drosophila suzukii species group" has been editorially accepted for publication in PLOS Genetics. Congratulations!

Yours sincerely,

Emily L Behrman

Guest Editor

PLOS Genetics

Justin Fay

Section Editor

PLOS Genetics

Aimée Dudley

Editor-in-Chief

PLOS Genetics

Anne Goriely

Editor-in-Chief

PLOS Genetics

BlueSky: @plos.bsky.social

Comments from the reviewers (if applicable):

**Data Deposition**

http://datadryad.org/submit?journalID=pgenetics&manu=PGENETICS-D-25-00456R1

**Press Queries**

---

## [Editor Report · Acceptance letter]

PGENETICS-D-25-00456R1

Molecular evolution of CO2-sensing ab1C neurons underlies divergent sensory responses in the Drosophila suzukii species group

Dear Dr Zhao,

We are pleased to inform you that your manuscript entitled "

Molecular evolution of CO2-sensing ab1C neurons underlies divergent sensory responses in the Drosophila suzukii species group" has been formally accepted for publication in PLOS Genetics! Your manuscript is now with our production department and you will be notified of the publication date in due course.

With kind regards,

Anita Estes

PLOS Genetics

On behalf of:
